



# Calving Fronts and Where to Find Them: A Benchmark Dataset and Methodology for Automatic Glacier Calving Front Extraction from SAR Imagery

Nora Gourmelon[1], Thorsten Seehaus[2], Matthias Braun[2], Andreas Maier[1], and Vincent Christlein[1]

[1]Pattern Recognition Lab, Friedrich-Alexander-Universität Erlangen-Nürnberg, Erlangen, Germany
[2]Institut für Geographie, Friedrich-Alexander-Universität Erlangen-Nürnberg, Erlangen, Germany

**Correspondence:** Nora Gourmelon (nora.gourmelon@fau.de)

**Abstract.** Exact information on calving front positions of marine- or lake-terminating glaciers is a fundamental glacier variable for analyzing ongoing glacier change processes and assessing other variables like frontal ablation rates. In recent years, researchers started implementing algorithms that could automatically detect the calving fronts on satellite imagery. Most studies use optical images, as in these images, calving fronts are often easy to distinguish due to sufficient spatial resolution and the

presence of different spectral bands, allowing the separation of ice features. However, detecting calving fronts on SAR images is highly desirable, as SAR images can also be acquired during the polar night and are independent of weather conditions, e. g., cloud cover, facilitating all-year monitoring worldwide. In this paper, we present a benchmark dataset of SAR images from multiple regions of the globe with corresponding manually defined labels to train and test approaches for the detection of glacier calving fronts. The dataset is the first to provide long-term glacier calving front information from multi-mission data.

As the dataset includes glaciers from Antarctica, Greenland and Alaska, the wide applicability of models trained and tested on this dataset is ensured. The test set is independent of the training set so that the generalization capabilities of the models can be evaluated. We provide two sets of labels: one binary segmentation label to discern the calving front from the background and one for multi-class segmentation of different landscape classes. Unlike other calving front datasets, the presented dataset contains not only the labels but also the corresponding preprocessed and geo-referenced SAR images as PNG files. The ease

of access to the dataset will allow scientists from other fields, such as data science, to contribute their expertise. With this benchmark dataset, we enable comparability between different front detection algorithms and improve the reproducibility of front detection studies. Moreover, we present one baseline model for each kind of label type. Both models are based on the U-Net, one of the most popular deep learning segmentation architectures. Additionally, we introduce Atrous Spatial Pyramid Pooling to the bottleneck layer. In the following two post-processing procedures, the segmentation results are converted into

one-pixel-wide front delineations. By providing both types of labels, both approaches can be used to address the problem. To assess the performance of the models, we first review the segmentation results using the recall, precision, $F_1$-score, and the Jaccard Index. Second, we evaluate the front delineation by calculating the mean distance error to the labeled front. The presented vanilla models provide a baseline of $150\,\text{m} \pm 24\,\text{m}$ mean distance error for the Mapple Glacier in Antarctica and $840\,\text{m} \pm 84\,\text{m}$ for the Columbia Glacier in Alaska, which has a more complex calving front, consisting of multiple sections, as

compared to a laterally well constrained, single calving front of Mapple Glacier.





# 1 Introduction

Ice mass loss of the ice sheets and glaciers is one of the major contributors to current global sea-level rise (Frederikse et al., 2020; Zemp et al., 2019; Sheperd et al., 2018; Khan et al., 2015). Alongside surface mass balance, ice losses at the calving fronts of marine- or lake-terminating glaciers are the major causes for ice depletion (Khan et al., 2015; Sheperd et al., 2018).

Hence, frontal ablation, defined as glacier mass loss due to calving and submarine melt (Dryak and Enderlin, 2020), is an essential parameter of total glacier mass balances. This is all the more true because ice discharge at the calving fronts of marine-terminating glaciers is a self-reinforcing system, as calving events lead to an expansion of the ice-melange triggered by the jamming wave and the break-up and thinning of the melange subsequently weakens the buttressing forces stabilizing the glacier (Robel, 2017). There exist some estimates of frontal ablation at glaciers outside the large ice sheets (Burgess et al.,

2013; McNabb et al., 2015; Minowa et al., 2021). Neglecting the frontal ablation can significantly bias the numerical glacier models. For example Recinos et al. (2019) reported an underestimation of ice thickness in Alaska in the range of 19 % on regional and up to 30 % on glacier scales when omitting frontal ablation. A spatio-temporal quantification of frontal ablation can therefore provide much needed reference data for glacier model parametrization (Recinos et al., 2019, 2021). Frontal ablation has successfully been parametrized for individual glaciers (Åström et al., 2014; Ultee and Bassis, 2016; Todd and

Christoffersen, 2014; Nick et al., 2010). These models, however, rely on data that are hard to obtain for entire glaciated regions (Recinos et al., 2019). An automated process to determine the frontal ablation based on easily accessible data is required. Changes in the position of the glaciers' calving fronts are crucial information for estimating frontal ablation. Satellite imagery facilitates the mapping of calving front positions in remote areas and on large spatial scales. Calving front positions can be acquired from optical as well as synthetic aperture radar (SAR) satellite imagery. Both imaging modalities have advantages

and disadvantages for calving front delineation (Baumhoer et al., 2018). Optical satellite imagery has higher spatial accuracy and often higher resolution compared to SAR imagery. Moreover, different spectral bands allow the separation of ice types in optical images (Tedesco, 2014; Gao et al., 1998), whereas the backscatter in SAR images might be similar for different ice types. The water–ice boundary in SAR images has, however, a high contrast (Liu and Jezek, 2004). Additionally, SAR can penetrate clouds and thin snow cover and is independent of solar illumination, allowing recordings during the night and polar

night. This leads to higher temporal availability, in particular at polar regions, for SAR images than for optical images. The majority of the calving glaciers are situated in polar regions (Hugonnet et al., 2021). Consequently, we decided to use SAR imagery for calving front delineation due to the better temporal coverage. Until recently, ice-shelf fronts and glacier termini in related work were usually manually delineated (Baumhoer et al., 2018). However, this approach is no longer feasible with the rapidly growing satellite image archives, as manual delineation is a highly time-consuming, tedious, and expensive task.

Recent studies (Baumhoer et al., 2019; Cheng et al., 2021; Davari et al., 2022, 2021; Hartmann et al., 2021; Heidler et al., 2021; Holzmann et al., 2021; Marochov et al., 2021; Mohajerani et al., 2019; Periyasamy et al., 2022; Zhang et al., 2019, 2021) focus on deep learning to extract the calving front and show great success. However, the evaluations of these studies were generally based on different datasets and are therefore not comparable. To bridge this gap, we introduce a benchmark dataset for calving





front delineation of marine-terminating glaciers located in the Arctic, Greenland, and Antarctica. It is the first dataset to provide
long-term calving front information from multiple missions.

To automatically delineate the calving front in SAR images, we perform a segmentation of the SAR image. In segmentation, each pixel is assigned to a specific class. In related works, two different segmentation approaches are used for the delineation. The first approach performs a binary segmentation between land and ocean (including ice-melange) and extracts the calving front from the boundary between the two classes in a post-processing step. The second approach directly assigns the pixels of the SAR image to the classes calving front and background. Both segmentation approaches are supervised learning tasks. Hence, ground truth segmentation images are needed for the algorithm to learn to distinguish the classes. As the classes in the two approaches differ (front and background versus land and ocean), different ground truth masks are needed for the two segmentation tasks. Likewise, our presented benchmark dataset has two ground truth annotations for each SAR image, one showing the segmentation front versus the background and the other one showing a segmentation into the classes ocean (including ice-melange), rock, glacier, and "no information available" (from now on called "NA"), which consists of SAR shadows, layover regions, and areas outside the swath. The second set does not represent a binary pixel classification task but a multi-class segmentation task. The calving front delineation is then carried out during post-processing using the boundary between the classes glacier and ocean.

In this study, we present the first publicly available annotated dataset for quantifying the performance of glacier calving front delineation algorithms on SAR images. Besides the calving front labels, the dataset also contains the corresponding preprocessed and geo-referenced SAR images enabling a direct application of deep learning segmentation models. Our train and test datasets include both images of where the ocean in front of the glacier front is covered by ice-melange and images where mainly open water is present. This ensures the generalizability of tested algorithms to all ocean surface settings. Two vanilla models are presented: one for the multi-class segmentation into glacier, ocean, rock outcrop and NA areas and one for the binary segmentation into front and background. Future models can use these models as a basis and compare against their performance on the presented dataset. For the evaluation of the algorithms, metrics are introduced that assess the segmentation performance of the presented Convolutional Neural Network (CNN) models and the result of the front delineation after post-processing.

The next section of this paper gives an overview of related work, including other datasets and algorithms for front delineation. In Sect. 3, we explain the details of our own dataset, while Sect. 4 introduces the baseline methods. The evaluation of these methods on our dataset is given in Sect. 5 before we draw some conclusions in Sect. 6.

## 2 Related work

Due to the high demand for information on the position of ice shelf fronts and glacier termini, several related datasets have been published and studies have been carried out aiming at automatically extracting these positions from satellite imagery.



## 2.1 Datasets

Existing datasets can be divided into SAR and optical imagery, as well as datasets that are based on both imaging modalities. Most datasets were constructed using solely optical images (Lippl, 2019; King and Howat, 2020; Fausto et al., 2019; Schild and Hamilton, 2013; Cheng et al., 2020). Lippl (2019) contains manually delineated calving front locations throughout the James Ross Island that were extracted using panchromatic Landsat-8 data. The remaining optical datasets feature delineations of Greenlandic glaciers. Calving front positions of Helheim and Kangerdlugssuaq are given by Schild and Hamilton (2013) based on MODIS imagery. Fausto et al. (2019) delineate 47 of the largest outlet glaciers in Greenland annually at the end of the melt season between 1999 and 2018 based on ASTER and Landsat imagery. Front position changes of an even larger number (234) of Greenlandic glaciers are displayed over more than three decades by King and Howat (2020). Their delineations are based on ASTER and Landsat 4-8 imagery. Another extensive database (Cheng et al., 2020) comprises 22,678 calving front lines across Greenlandic marine-terminating glaciers. Parts of the calving fronts were manually delineated using optical Landsat NIR band imagery while the deep learning algorithm CALFIN (Cheng et al., 2021) produced the other part of the provided calving fronts. CALFIN was trained and tested on the manually mapped fronts included in the dataset as well as additional SAR images from Antarctica. Only two datasets were constructed using solely SAR imagery, which are both located at Jakobshavn Isbræ in Greenland. Zhang (2019a) comprises manually delineated calving fronts based on TerraSAR-X imagery that Zhang et al. (2019) used to train a neural network to extract calving fronts from SAR images automatically, while Zhang (2019b) includes the calving fronts that the trained network extracted from additional TerraSAR-X images. The ESA Greenland Ice Sheet CCI project team (2019) provides calving front positions of 28 major outlet glaciers in Greenland. The manual delineation is carried out based on both SAR and optical imagery taken from ERS-1/2, Envisat, Sentinel-1 and Lansat 5, 7, and 8. The dataset is continuously updated; we refer to version 3 here. Optical and SAR imagery from Landsat-8, Sentinel-2, Envisat, ALOS-1, TerraSAR-X, Sentinel-1, and ALOS-2 was also used by Zhang et al. (2020a) to manually delineate Jakobshavn Isbræ, Kangerlussuaq and Helheim glaciers. This dataset was again used to train and test a deep learning network that was afterwards used to produce a second dataset (Zhang et al., 2020b). Two massive databases that need to be mentioned for the sake of completeness are the Global Land Ice Measurements from Space (GLIMS) (Raup et al., 2018) and the SCAR Antarctic Digital Database (ADD) (Gerrish et al., 2021). ADD includes Antarctica's coastline south of 60°S including grounding lines and ice shelf fronts. Up to now, GLIMS provides 546,300 glacier outlines from glaciers around the world with an ingest rate of 7529 outlines per month. Their outlines are based on optical imagery.

An overview of all datasets can be seen in Table 1. Every described dataset only provides the glacier outlines or polygons and not the corresponding preprocessed satellite imagery, hence, hindering the direct application of deep learning algorithms.

## 2.2 Algorithms

Since 2019, several studies have applied deep learning techniques to delineate the calving front of marine-terminating glaciers in satellite imagery. The first studies are all based on the U-Net (Ronneberger et al., 2015), which still is the basis of many state-of-the-art networks in image segmentation. Mohajerani et al. (2019) employs this encoder-decoder network to segment



| Modality | Dataset | Annotation | Area | # Glaciers | # Mapped Fronts | Time Span | Res. |
|---|---|---|---|---|---|---|---|
| Optical | [Lippl] | Manually | Antarctica | 26 | 656 | 2014 - 2018 | 15 m |
| | [King] | Manually | Greenland | 234 | 128,442 | 1985 - 2018 | 30 m |
| | [Fausto] | Manually | Greenland | 47 | 1180 | 1999 - 2018 | 10 - 30 m |
| | [Schild] | Manually | Greenland | 2 | 1862 | 2001 - 2010 | 250 m |
| | [Cheng] | Manually & Network | Greenland | 66 | 22,678 | 1972 - 2019 | 30 m |
| | [ADD] | Manually & Semi-Automatic | Antarctica | | | Since 1843 | |
| | [GLIMS] | Manually & Semi-Automatic | Global | ∼200,000 | 546,300 | Since 1750 | |
| SAR | [Zhang19a] | Manually | Greenland | 1 | 159 | 2009 - 2015 | 3 m |
| | [Zhang19b] | Network | Greenland | 1 | 159 | 2009 - 2015 | 3 m |
| | *Ours* | Manually | Alaska, Antarctica & Greenland | 7 | 681 | 1996 - 2020 | 6 - 20 m |
| Optical & SAR | [Zhang20a] | Manually | Greenland | 3 | 2087 | 2002 - 2019 | 3 - 40 m |
| | [Zhang20b] | Network | Greenland | 3 | 2087 | 2002 - 2019 | 3 - 40 m |
| | [ESA] | Manually | Greenland | 28 | 1089 | 1990 - 2016 | 10 - 30 m |

**Table 1.** Overview of publicly available front line datasets. The *Annotation* entry refers to how the delineations were produced, i. e., manually, by a network or by a model. The entry *# Glaciers* gives the number of presented glaciers and *# Mapped Fronts* the number of glacier front delineations over all glaciers inherent in a dataset. The *Res.* indicates the spatial resolution of images used for the mapping of the glacier fronts. The datasets are: [Lippl] Lippl (2019), [King] King and Howat (2020), [Fausto] Fausto et al. (2019), [Schild] Schild and Hamilton (2013), [Cheng] Cheng et al. (2020), [ADD] (Gerrish et al., 2021), [GLIMS] (Raup et al., 2018), [Zhang19a] Zhang (2019a), [Zhang19b] Zhang (2019b), [Zhang20a] Zhang et al. (2020a), [Zhang20b] Zhang et al. (2020b), [ESA] ESA Greenland Ice Sheet CCI project team (2019), and the dataset presented in this paper (Gourmelon et al., 2022b) is denoted as *Ours*.





multispectral Landsat images into calving front and background. Baumhoer et al. (2019) and Zhang et al. (2019) performed the first analyzes based on SAR imagery. Zhang et al. (2019) employ Zhang (2019a)'s dataset for training and testing, while

Baumhoer et al. (2019)'s database is not publicly available. In a subsequent study (Baumhoer et al., 2021), Baumhoer et al. modified their U-Net and used it to delineate the entire Antarctic coastline for 2018.

    The U-Net architecture was replaced with DeepLabv3 (Chen et al., 2018b) by both Zhang et al. (2021) and Cheng et al. (2021). Both studies segment optical and SAR imagery into land and sea, including sea ice. Zhang et al. (2021) performs a comparison between the U-Net and DeepLabv3+ with different backbones. They published their manual delineations in

Zhang et al. (2020a). In contrast to Zhang et al. (2019), Cheng et al. (2021) employs the Xception model (Chollet, 2017) as backbone like in the original DeepLabv3 paper (Chen et al., 2018b). Cheng et al. (2021)'s network, called CALFIN, outputs two probability masks: land versus sea and coastline versus background. They evaluate CALFIN not only on their own published dataset (Cheng et al., 2020) but also on images from Mohajerani et al. (2019), Zhang et al. (2019), and Baumhoer et al. (2019). Another study that learns two tasks simultaneously is conducted by Heidler et al. (2021). Their network is based on the U-Net

architecture but features two output heads: one for the segmentation into sea and land and one for delineating the coastline. They compare their model against other deep learning approaches including the U-Net model from Baumhoer et al. (2019). While their code is open source, their dataset is not publicly available. A completely different approach to front delineation is taken by Marochov et al. (2021). Instead of directly segmenting the entire images into the desired classes, Marochov et al. (2021) use classification networks to determine the class of every single pixel in each image separately. In their paper, Marochov et al.

(2021) compare their model's results with the results given in Baumhoer et al. (2019)'s, Cheng et al. (2021)'s, Mohajerani et al. (2019)'s and Zhang et al. (2019)'s studies. However, this is not a valid comparison because the models were trained and tested on different datasets, so the differences in performance can not solely be attributed to the models but may be due to the data itself. Five more studies perform their experiments on SAR imagery solely. Holzmann et al. (2021) incorporate attention gates in the skip connections of the U-Net to improve the segmentation performance. Hartmann et al. (2021) implement a Bayesian

U-Net to quantify the uncertainty of the model and use the uncertainty as an additional input to a second U-Net to improve its prediction of the calving front position. Davari et al. (2021) use a distance map-based loss to train their U-Net. Periyasamy et al. (2022) further optimize the feature extraction of the U-Net. Lastly, Davari et al. (2022) formulate the front segmentation as a regression task, letting a U-Net predict the distance of each pixel to the front. All five studies, use Zhang et al. (2019)'s U-Net as baseline, but do not publish their dataset.

About half of the presented studies make their code publicly available (Cheng et al., 2021; Davari et al., 2021; Heidler et al., 2021; Marochov et al., 2021; Mohajerani et al., 2019; Zhang et al., 2021). Only two-thirds of the described studies make an effort to compare their method with other existing methods by either testing on the same dataset or by re-training and testing the other's model on the own dataset (Cheng et al., 2021; Davari et al., 2022, 2021; Hartmann et al., 2021; Heidler et al., 2021; Holzmann et al., 2021; Periyasamy et al., 2022; Zhang et al., 2021). Furthermore, testing on another's dataset alone without

re-training the own model on the other's training dataset beforehand is not a strong comparison, as the own training dataset might form a better learning basis. The low rate of informative comparisons is understandable, as most datasets and codes are not easily accessible or employable. Satellite imagery is not provided, and preprocessing steps like geo-referencing need to be





repeated. Therefore, a benchmark dataset and an easily reusable baseline code are highly needed. The dataset and code will benefit not only the comparability of future studies but also their reproducibility and the broad applicability of future models.

**3 Data set**

**3.1 Study sites**

Seven tidewater glaciers situated in Greenland, Alaska and on the Antarctic Peninsula (AP) were selected to generate labels to train and evaluate automated calving front mapping approaches for SAR imagery (see Fig. 1). The glaciers were selected considering existing data sets of calving fronts, available SAR coverage and reports on calving front variability.

Along the AP, five former ice-shelf tributary glaciers were selected. The AP is a hot spot of global warming, and a significant temperature increase was observed during the 20th century (Oliva et al., 2017; Turner et al., 2016). As a result, Cook and Vaughan (2010) reported rapid retreat and even disintegration of various ice shelves along the AP. In 1995, the Prince-Gustav-Channel and Larsen-A ice shelves broke up, followed by the disintegration of the Larsen-B Ice Shelf in 2002 (Cooper, 1997; Scambos et al., 2004; Skvarca et al., 1998). Consequently, the former ice shelf tributary glaciers reacted with increased ice

discharge and further frontal recession due to the loss of buttressing forces by the ice shelves (e.g., Rott et al., 2014; Seehaus et al., 2015, 2016). For this benchmark database, we selected the Dinsmoore-Bombardier-Edgworth (DBE) and Sjögren-Inlet (SI) glacier systems, which were major tributaries of the Larsen-A and Prince-Gustav-Channel ice shelves, respectively. At the Larsen-B embayment, the former ice shelf tributaries Crane, Mapple and Jorum were chosen. Similar reaction patterns were observed at these glaciers. A significant rise in ice flux after the ice shelve break-ups, followed by a long-term decrease, was

measured. Concurrently, the glacier fronts retreated strongly after the disintegration of the ice shelf and partially stabilized or showed a readvance again after a few years (e.g., Rott et al., 2014, 2018; Wuite et al., 2015; Seehaus et al., 2015, 2016).

At Greenland, we incorporated Jakobshavn Isbrae (JAC) in our database. It is located on the west coast and drains the Greenland ice sheet. For the last decades, pronounced ice flow and calving front variabilities were reported (Joughin et al., 2008, 2012). A frontal retreat of 16 km between 2002 and 2008 was revealed by Rosenau et al. (2013). Correlations between

changes in the calving front positions and variations in ice discharge and the formation of ice melange in the glacier fjord were observed by various analyses (Amundson et al., 2010; Joughin et al., 2008, 2012).

In Alaska, we selected Columbia Glacier. It is a large marine-terminating glacier and has strongly retreated since the early 1980s (McNabb et al., 2015; Krimmel, 2001)). It has split into two branches - the main and the west branch in 2010 (Vijay and Braun, 2017). Between 1957 and 2007, an average ice thinning rate of 10 m/a was found (McNabb et al., 2012), and

pronounced seasonal and interannual variability of the ice flow and calving front position was reported by Vijay and Braun (2017).

At all selected glaciers, there exists a good temporal coverage by SAR imagery. In particular high temporal coverage by TerraSAR-X and TanDEM-X strip map imagery exists since most glaciers are part of the so-called TanDEM-X super-test-sites. For DBE and SI glacier systems, a detailed analysis of the calving front evolution, based on manual picking of the front

**Figure 1.** Overview of sampled glaciers in Alaska, Greenland and on the Antarctic Peninsula. Pink polygons highlight the subset areas used for the data generation. Background: ESRI Satellite ©ESRI



positions on multi-mission SAR imagery, exists (Seehaus et al., 2015, 2016) and was incorporated in this benchmark database. At the other glaciers, the glacier fronts were as well manually picked on SAR intensity imagery.

## 3.2 Data set generation

We used SAR imagery from the satellite missions ERS-1/2, Envisat, RADARSAT-1, ALOS PALSAR, TerraSAR-X (TSX), TanDEM-X (TDX), and Sentinel-1, covering the period 1995-2020. The SAR data was provided by the German Aerospace
Center (DLR), the European Space Agency (ESA) and the Alaska Satellite Facility (ASF). The SAR imagery was provided as single-look-complex (SLC) data, except for some RADARSAT-1 imagery on the AP, which was provided in Precision Image (PRI) format (similar to a multi-looked intensity image). At first, the SAR images were calibrated and multi-looked to obtain approximately squared pixel sizes and to reduce speckle noise. The applied multi-looking factors are provided in Table 2. They were selected based on a trade-off between loss of spatial resolution and speckle noise reduction. Subsequently, the SAR
intensity imagery was geocoded and orthorectified. On the AP, the enhanced ASTER digital elevation model (DEM) of the AP (Cook et al., 2012) was employed, whereas for Columbia Glacier, the Shuttle Radar Topography Mission (SRTM) DEM and for Jakobshavn Glacier, the global TanDEM-X DEM were used. The specifications and parameters of the SAR sensors and imagery are provided in Table 2.

| Platform | Sensor | Mode | SAR band | Repetition cycle [d] | Time interval | Multi looking factor | Approx. slant range × azimuth res. [m] | Ground range res. [m] |
|---|---|---|---|---|---|---|---|---|
| ERS-1/2 | SAR | IM | C band | 35/1 | 13. November 1995 - 02. April 2010 | 1×5 | 8×4 | 20 |
| RADARSAT 1 | SAR | ST | C band | 24 | 10. September 2000 - 20. January 2008 | 1×4 | 12×5 (SLC) | 20 (SLC) 12.5 (PRI) |
| Envisat | SAR | IM | C band | 35 | 05. December 2003 - 03. July 2010 | 1×5 | 8×4 | 20 |
| ALOS | PALSAR | FBS | L band | 46 | 18. May 2006 - 03. March 2011 | 2×5 | 5×4 | 16.7 |
| TerraSAR-X TanDEM-X | SAR | SM | X band | 11 | 13. October 2008 - 20. May 2016 | 3×3 | 1.4×2 | 6.7 |
| Sentinel-1A/B | SAR | IW | C band | 6/12 | 18. December 2015 - 12. June 2020 | 5×1 | 4×20 | 20 |

**Table 2.** Summary of the SAR satellites and specifications of the used imagery. All imagery was provided in SLC format; only for RADARSAT-1, some data takes were provided in PRI format.





The SAR data processing was done using the GAMMA RS Software. The manually picked glacier front positions employed
for the studies by Seehaus et al. (2015, 2016) were used for DBE and SI glacier systems. Additionally, glacier front positions
were manually mapped at the Larsen-B embayment and for Columbia Glacier. At Jakobshavn Isbrea, Zhang et al. provided
only spatial lines of their manually picked calving front positions. Thus, we ordered TSX/TDX SLC strip map imagery at
DLR from the same orbits and dates as used by Zhang et al. (2019) and applied the same SAR processing as mentioned above
in order to have the same imagery setting for each satellite at all of our study sites. Since Zhang et al. applied an additional
geo-referencing step using manually defined ground control points by means of Google Earth imagery, which we could not
completely replicate, there was still a spatial offset between our SAR imagery and their front positions. Thus, we manually
re-mapped all glacier fronts at Jakobshavn Isbrea. A quality factor ranging from 1 to 6 was assigned to each calving position.
The quality factors are a subjective measure of the reliability of the picked front position depending on the similarity of the ice
melange and the glacier. Table 3 shows the quality factors and the respective uncertainty values perpendicular to the glacier
front.

| Quality factor | 1 | 2 | 3 | 4 | 5 | 6 |
|---|---|---|---|---|---|---|
| [m] | 70 | 130 | 200 | 230 | 450 | >450 |

**Table 3.** Quality factors of manual calving front mapping and the related horizontal uncertainty of the mapped glacier front position.


Next, the geo-referenced SAR intensity imagery of all glaciers was cropped to the areas of interest (see Fig. 1) and converted
to 16 bit single-channel images. In order to define different surface types in the SAR imagery, the manually mapped calving
fronts (spatial line) were combined with a stable glacier outline data set. At the AP, the "ice feature catchments" and "rock-
outcrop" polygons from ADD were taken. At Jakobshavn Isbrea, we manually generated the relevant layers. At Columbia
Glacier, we relied on the Randolph Glacier Inventory 6.0, which was slightly manually refined since Columbia Glacier retreated
and thinned strongly and new rock outcrops were formed, which are not included in the inventory. 8 bit single-channel images
identifying four *zones* – "ocean", "rock outcrops", "glacier area" and "no information available" (SAR shadow, layover regions
and areas outside the SAR swath) in the SAR imagery – were generated based on these spatial features. Additionally, 2-
class imagery was generated using the picked calving front locations solely. The categorized images have the same extent,
dimensions and spatial resolution as the respective SAR image subsets, allowing the definition of two sets of labels for the
SAR image subsets that can be used for machine learning methods: one binary set for direct front segmentation; and one
set, including the 4-class images, for multi-class "zones" segmentation. To support the post-processing of the calving front
detection, we provide additional bounding box information on the maximum glacier front extents at each glacier for each label
(see Sect. 4.3).
The resulting database, comprising the preprocessed SAR image subsets and both label sets, was split into train and test
samples. The samples from DBE, SI, Crane, Jorum glaciers and JAC are used to train the calving front detection algorithms.
This training data set comprises samples of different glacier geometries (see Fig. 1) and data from all incorporated SAR



sensors. The samples from Mapple and Columbia glaciers are used to evaluate the performance of the neural networks. At both glaciers, the samples contain imagery with open ocean and ice-covered sea surface next to the calving front. Mapple Glacier

has a relatively simple glacier geometry, comprising a single calving front, well constrained by the fjord valley walls. The geometry of Columbia Glacier is much more complex. It consists of multiple branches, and the strong glacier retreat causes the split of the calving front into various sections. The number of images per glacier, sensor and label set is summarized in Table 4.

| Platform | DBE | SI | Jorum | Crane | Mapple | Columbia | Jacobshavn | Training | Test | All |
|---|---|---|---|---|---|---|---|---|---|---|
| ERS-1/2 | 16 | 28 | 3 | 5 | 2 | | | 52 | 2 | 54 |
| RADARSAT | 27 | 27 | | | | | | 54 | | 54 |
| Envisat | 26 | 27 | 10 | 9 | 10 | | | 72 | 10 | 82 |
| ALOS | 20 | 7 | 6 | 7 | 8 | | | 40 | 8 | 48 |
| TerraSAR-X TanDEM-X | 44 | 32 | 43 | 48 | 22 | 47 | 159 | 326 | 69 | 395 |
| Sentinel-1A/B | | | 15 | | 15 | 18 | | 15 | 33 | 48 |
| Sum | 133 | 121 | 77 | 69 | 57 | 65 | 159 | 559 | 122 | 681 |

**Table 4.** Summary of labels for each glacier and sensor, as well as for training and test data sets.

## 4 Baseline methods

Alongside our benchmark dataset, we present two models, one for the direct front segmentation and one for the zone segmentation, which can serve as baseline methods for future studies. The complete workflow comprises preprocessing, processing with the neural network and post-processing, including the front extraction. An overview of this workflow is given in Fig. 2.

### 4.1 Preprocessing

Since most of the standard preprocessing has already been done for the benchmark dataset (see Sect. 3), only preprocessing

techniques related to the specific architecture of the neural network need to be applied. In the front segmentation model, we thicken the front labels by morphological dilation employing a rectangular structuring element of size $5 \times 5$ pixels. Thickening the front mitigates the class imbalance inherent in the calving front dataset. A problem due to class imbalance arises when the class distributions are highly imbalanced, leading to low prediction accuracy for the rare class (Ling and Sheng, 2010). In our case, the front is only a one-pixel-wide line and therefore has much fewer pixels than the background. Hence, we use this

morphological dilation and a specialized loss function for the training, which is described in Sect. 5.2. For the evaluation, the one-pixel-wide fronts are used.



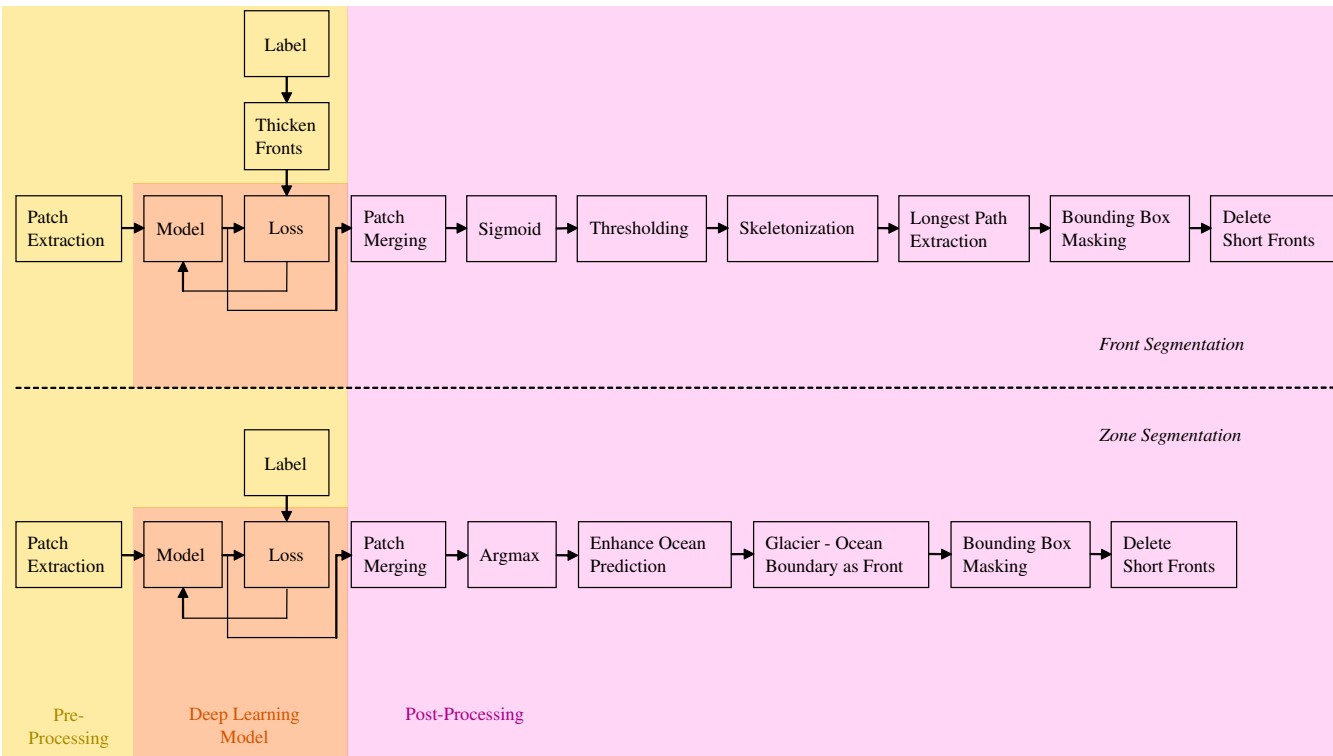

**Figure 2.** Workflow for both front and zone segmentation. The enhancement of the ocean prediction includes filling gaps in the ocean zone and selecting only the largest connected ocean component as the ocean region.

Another preprocessing step that we apply is patch extraction, where each image is divided into tiles of the same size. Neural networks usually process several images in one forward pass. This group of images is called a batch, and a batch size of more than one allows for faster training (Bengio, 2012). Since the images in our dataset are large and have different sizes, there are

two ways to achieve a sufficient batch size that still fits in the GPU: resizing and patch extraction. We choose patch extraction, as during resizing, information is lost, while the downside to patch extraction is the loss of the global context of the patches. Before the images can be divided into patches, each image must be padded with zeros so that no remainder is left that is smaller than the size of the patch. Next, a sliding window approach (see Fig. 3) is used to extract patches of size $256 \times 256$ pixels. The step size for the validation and test images is set to 128 so that the patches overlap and that stitching artefacts can be reduced

when the images are reassembled, cf. Sect. 4.3. For training images, the stride is chosen such that the patches do not overlap, as this reduces the computational load considerably, and the network still sees the complete dataset once.

## 4.2 Deep learning models

Both our models are based on the U-Net (Ronneberger et al., 2015), an encoder-decoder CNN, which is used in most state-of-the-art networks for image segmentation. We adapt the standard U-Net architecture to our needs and refine it by inserting



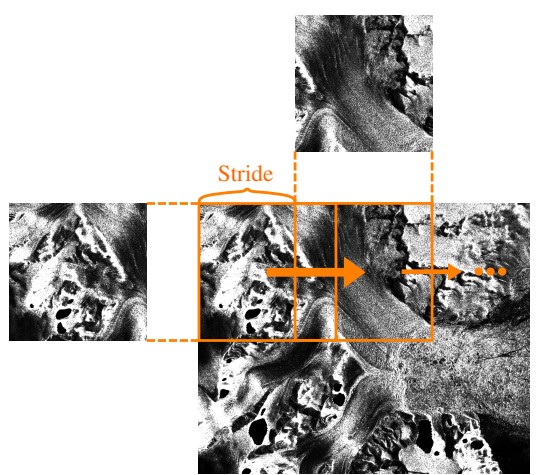

**Figure 3.** Sliding window approach for patch extraction. A window with the desired size of the patches is slid over the image with a certain stride or step size.

Atrous Spatial Pyramid Pooling (ASPP) (Chen et al., 2018a) in the bottleneck layer. ASPP applies atrous convolutions with different dilation rates in parallel, resulting in differently sized fields of view, and then fuses the obtained feature maps, allowing robust segmentation of objects at multiple scales. Since calving fronts, like any other geologic structure, do not have a fixed size, this specialized layer is advantageous for our models. Our chosen dilation rates are 1, 2, 4, and 8. Other changes to the original U-Net architecture are the choice of Leaky ReLU with a negative slope of 0.1 as activation layer, 32 as the number of start feature layers, and "same" instead of "valid" padding. Our complete architecture can be seen in Fig. 4. The output of the model depends on the type of segmentation: In direct front segmentation, we receive a probability map where each pixel value indicates the predicted probability (however not normalized to $\in [0, 1]$ – the sigmoid function still needs to be applied) that this pixel belongs to the front. In zone segmentation, we get such a probability map for each zone. For the delineation of the calving front, these probability maps are further processed during post-processing.

## 4.3 Post-processing for front delineation

During post-processing, the output probability patches are first merged into complete images. For merging, we do not simply use the average of the values where the patches overlap but apply Gaussian importance weighting before taking the average, which reduces stitching artefacts. Gaussian importance weighting gives higher weights to pixels close to the center of patches and lower weights to pixels near the edge of the patches (Isensee et al., 2021). Afterwards, the area padded with zeros during preprocessing in the SAR image is removed from our output.



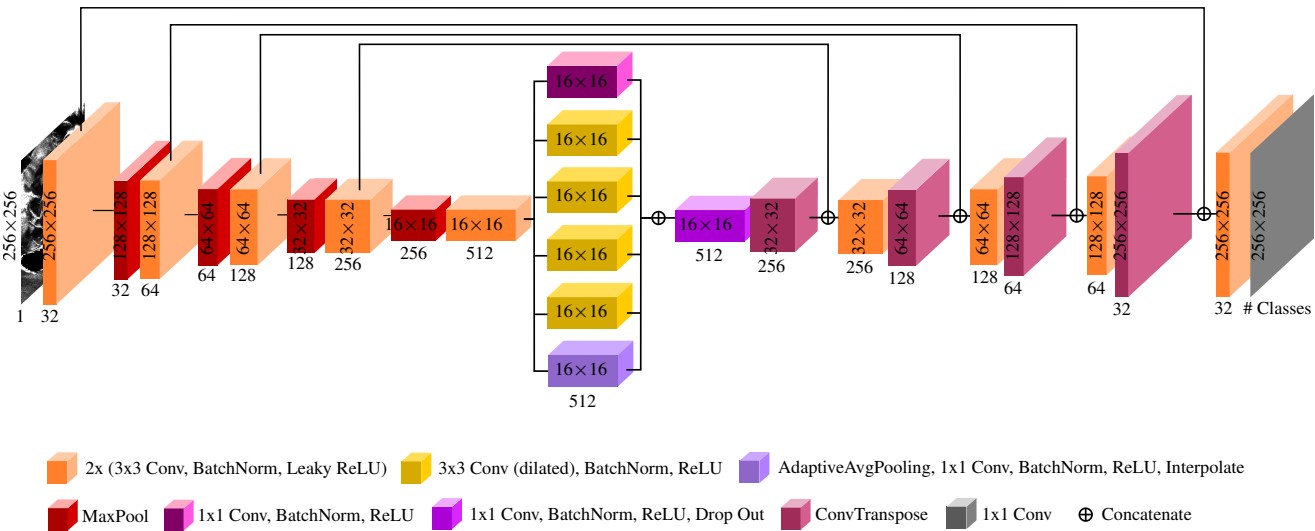

**Figure 4.** The architecture of both presented segmentation models. Different blocks denote the output of the transformations indicated by the color of each block. Note that the sizes of the blocks are not true to scale. The numbers beneath the blocks indicate the number of feature maps. The size of the features maps is indicated inside or next to the blocks. The number of classes given by *# Classes* is one for front segmentation and four for zone segmentation.

### 4.3.1 Zone segmentation

For zone segmentation, we obtain the zone prediction for each pixel by looking at all the probability maps and selecting the zone whose probability map has the highest value at the given pixel. To fill gaps in the ocean zone, which is required for further steps, we do not apply morphological filtering like Baumhoer et al. (2019, 2021). Instead, we perform a Connected Component
Analysis (CCA) on the inverted predicted ocean region to receive all connected non-ocean regions and mark all but the largest connected component in the original prediction to belong to the ocean region. In this way, we can guarantee to fill small (and larger) gaps inside the ocean zone without altering the zone's outer boundary. This might eliminate islands (which were not present in our training data), which is, however, unimportant for the front delineation task. Performing CCA on the ocean zone and omitting all but the largest connected component leaves us with one ocean area. The boundary between this ocean area
and all adjacent predicted glacier zones yields the one-pixel-wide glacier termini. We mask the glacier termini with a bounding box specified separately for each glacier to obtain only the calving front of the observed dynamic glacier. Finally, independent front segments shorter than 750 m are removed because they likely belong to static ice-ocean boundaries that are only partially excluded by the bounding box and in which we are not interested in mapping. The shortest front in our dataset is 1.5 km long. Thus, we choose the minimum reasonable length of a predicted calving front to be half of this length. However, be aware that
this parameter might need to be adjusted for new glaciers.





### 4.3.2 Front segmentation

To receive the front prediction, we first apply the sigmoid function pixel-wise on the probability map to receive predicted probability values between 0 and 1. Next, we use a threshold of 0.12 to binarize the prediction. If the probability value is higher than the threshold, then the pixel belongs to the front and otherwise to the background. The obtained prediction is not
yet a one-pixel-wide line. Therefore, we use skeletonization to extract a network of one-pixel-wide lines from our prediction. As these lines still show branches, we search for the longest path in each separate skeleton and delete the remaining branches to obtain filaments. We employ the method of Koch and Rosolowsky (2015) to receive filaments from skeletons. As with zone segmentation, we use the specified bounding boxes to hide static ice-ocean interfaces and discard fronts shorter than 750 m.

## 5 Evaluation

### 5.1 Evaluation metrics

For different tasks, different evaluation metrics are needed. Therefore, this section presents appropriate evaluation metrics for zone and front segmentation and the main objective, front delineation. Segmentation evaluation metrics can only tell how well the segmentation itself works, not how close the resulting delineated front is to the actual front after post-processing. Hence, our segmentation metrics only assess how good the intermediate result, i. e., the segmentation, is. If the intermediate
result is not sufficient, post-processing will likely not fix the problem. Therefore, an evaluation of segmentation performance is also necessary, primarily since it directly evaluates the neural network's performance and can provide insight into whether performance problems lie with the neural network or with the post-processing of its results.

### 5.1.1 Segmentation

Performance metrics for supervised classification and segmentation tasks are calculated using the confusion matrix, which gives
the number of true positives, false positives, true negatives, and false negatives (Fawcett, 2006). In binary segmentation, like our front segmentation, the pixels that belong to the desired class are referred to as positive; the remaining pixels are referred to as negative. The prefixes "true" and "false" indicate whether the division into positive and negative pixels is consistent with the prediction stated in the label or contradicts the label. In multi-class segmentation, like our zone segmentation, each class is first evaluated separately by considering the other classes as background and thus negative. Separately evaluating each class
allows multi-class problems to be split into several binary tasks, and a separate confusion matrix can be determined for each class. Thus, a value for the metric is obtained for each class. To get a collective value for the metric, one can average the class-wise metrics. We use both the class-wise and multi-class metrics to evaluate the zone model because the individual class-wise metrics provide valuable insight into how well the predictions are for each class. The averaged multi-class metric gives us an overview of how well the segmentation is working in general, but only the class-wise metrics allow us to analyze individual
classes, e. g., that the rock outcrop segmentation is accurate, but the glacier segmentation is poor.



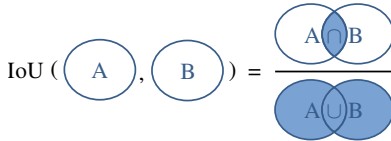

**Figure 5.** A visualization of the Jaccard Index or Intersection over Union (IoU). The two ellipses $A$ and $B$ signify two sets.

We apply four common metrics (Pedregosa et al., 2011): recall, precision, $F_1$-score, and the Jaccard Index (Jaccard, 1912) also called Intersection over Union (IoU). The recall is the percentage of all positive pixels predicted to be positive, while precision is the percentage of positively predicted pixels among all positive pixels (Lewis, 1990). The $F_1$-score is the harmonic mean of recall and precision (Fawcett, 2006). As the name says, the IoU divides the intersection of two sets by their union, where these two sets are the pixels of the predicted class and the pixels of the actual target for binary segmentation. A visual interpretation of the IoU is shown in Fig. 5. The equations 1 – 4 provide the calculations of the four metrics from true positives ($TP$), false positives ($FP$), true negatives ($TN$), and false negatives ($FN$). We calculate all metrics on the binarized predictions.

$$\text{Recall} = \frac{TP}{TP + FN} \tag{1}$$

$$\text{Precision} = \frac{TP}{TP + FP} \tag{2}$$

$$F_1 \text{ - Score} = 2 \cdot \frac{\text{Precision} \cdot \text{Recall}}{\text{Precision} + \text{Recall}} \tag{3}$$

$$\text{IoU} = \frac{TP}{TP + FP + FN} \tag{4}$$

### 5.1.2 Front delineation

As evaluation metric for the front delineation, we employ the mean distance error (MDE), which is adapted from Baumhoer et al. (2019), Cheng et al. (2021), Heidler et al. (2021), Mohajerani et al. (2019, 2021), and Zhang et al. (2019, 2021). Each predicted front is compared to its ground truth. The Euclidean distance to the closest pixel in the ground truth front is calculated for each pixel in the predicted front. To make the metric symmetric, the distance to the closest pixel in the predicted front is also calculated for each pixel of the ground truth front. These distances are stored for all images, and the average is taken over all images, forming the mean distance error. The analytical formula of the MDE follows as:

$$\text{MDE}(\mathcal{I}) = \frac{1}{\sum_{(\mathcal{P}, \mathcal{Q}) \in \mathcal{I}} (|\mathcal{P}| + |\mathcal{Q}|)} \sum_{(\mathcal{P}, \mathcal{Q}) \in I} \left( \sum_{\boldsymbol{p} \in \mathcal{P}} \min_{\boldsymbol{q} \in \mathcal{Q}} \|\boldsymbol{p} - \boldsymbol{q}\|_2 + \sum_{\boldsymbol{q} \in \mathcal{Q}} \min_{\boldsymbol{p} \in \mathcal{P}} \|\boldsymbol{p} - \boldsymbol{q}\|_2 \right) \tag{5}$$

with $\mathcal{I}$ being the set of all evaluated images, $\mathcal{P}$ the set of ground truth front pixels of one specific image, $\mathcal{Q}$ the set of corresponding predicted front pixels in that image, and $|.|$ denotes the cardinality of a set.





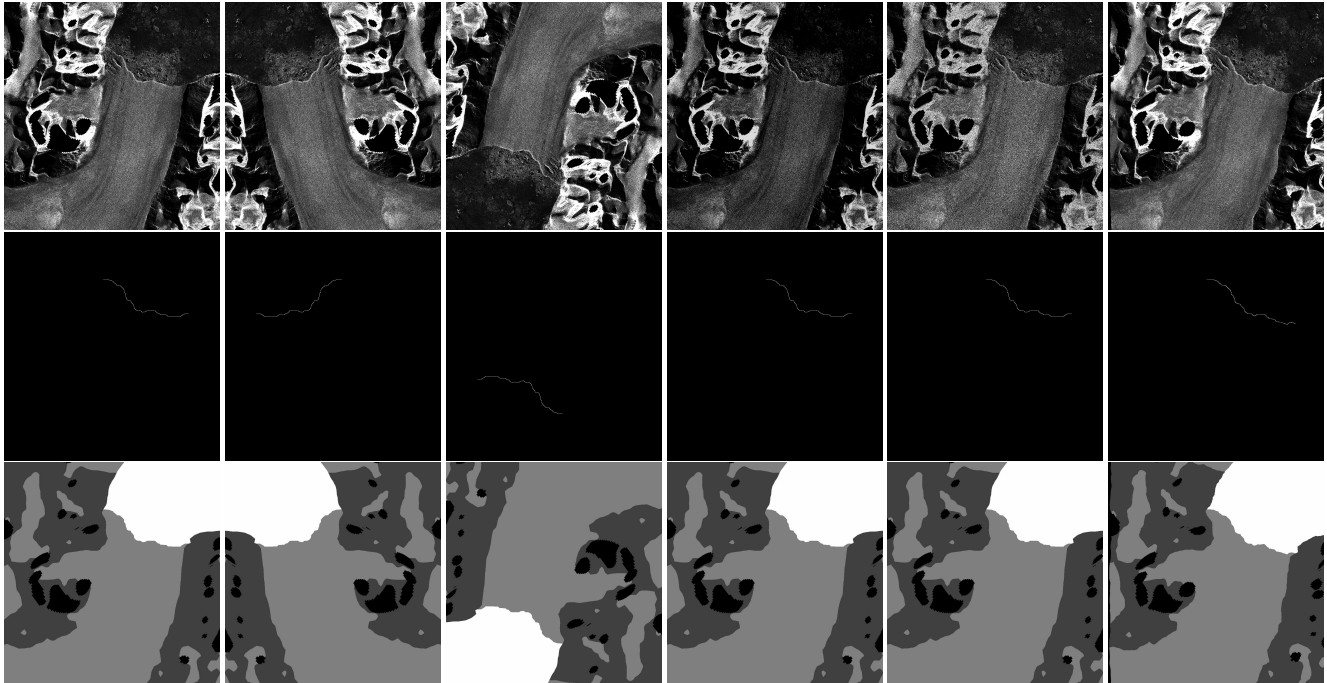

**Figure 6.** Applied augmentations. From left to right: Original image/label, horizontal flip, rotation (here $180°$), brightness adjustment, Gaussian noise and wrap. The first row shows the SAR images; the second one gives the label for the front segmentation, and the third one the label for the zone segmentation. For brightness adjustment, Gaussian noise and "wrap" augmentations, the intensities of the transforms have been enhanced such that they are better visible.

## 5.2 Experimental protocol

For our experiments, we further split the training dataset presented in Sect. 3 into a part used for training and a part used for validation. The ratio is nine to one for training versus validation. All hyperparameter optimizations were done, comparing the results on the validation set. Using the validation set allows us to frequently check the performance of our model during implementation while keeping the final evaluation on the test set unbiased. Like the test set, the validation set is preprocessed with overlapping patch extraction, which increases the amount of validation data relative to the training data. On-the-fly aug-
mentations are applied to the training set only. These random augmentations include rotations, horizontal flips, brightness adjustments, Gaussian noise, and a transform called "wrap" that acts like many simultaneous elastic transforms with Gaussian sigmas set at various harmonics (Nicolaou et al., 2022). A visualization of these augmentations is presented in Fig. 6. The probabilities with which these augmentations are applied to the input were determined empirically and depend on the segmentation task. For front segmentation, all augmentations are applied with a probability of 0.65; for zone segmentation, brightness
adjustment and wrap are applied with a probability of 0.1, Gaussian noise and rotation with 0.5, and flipping with a probability of 0.3.





Next, all patches (including validation and test set) are z-score normalized using the mean and standard deviation of the training dataset. The patches are fed into the neural network with a batch size of 16, as this was the maximum amount that could fit into the GPU using a patch size of $256 \times 256$. So there is a trade-off between patch size and batch size. We choose a

patch size of $256 \times 256$ to ensure a sufficiently large global context and adjust the batch size accordingly. To train the models, we employ a cyclic learning rate scheduler (Smith, 2017) in combination with the Adam optimizer (Kingma and Ba, 2015) and apply different loss functions for each segmentation task. For the zone segmentation model, the base learning rate for the scheduler is chosen to be $4 \cdot 10^{-5}$, and the maximum learning rate $2 \cdot 10^{-4}$. For the front segmentation model, the base learning rate is $1 \cdot 10^{-4}$, and the maximum learning rate is $5 \cdot 10^{-4}$. We use the rule of thumb given in (Smith, 2017) to select

a proper step size for both models. Hence, the step size is set to $30,000$, which roughly equals the number of training patches divided by the batch size times ten. Moreover, we perform gradient clipping to avoid exploding gradients. The global norm of the gradients, i.e., the norm calculated over all model parameters is truncated to 1.0 for both models. The values for gradient clipping, base, and maximum learning rates were determined empirically using the hyperparameter optimization framework optuna (Akiba et al., 2019). The loss function for zone segmentation is a weighted combination of Dice (Sudre et al., 2017)

and Cross-Entropy loss (Bishop, 1995). With optuna, the weighting was determined to be optimal if both parts of the combined loss function are weighted equally. For front segmentation, an improved distance map loss (Davari et al., 2021) is employed. Davari et al. (2021)'s improved distance map loss multiplies the network's prediction by weights calculated based on the front label before computing the cross-entropy of the prediction and the front label. We replace the cross-entropy with the Dice loss, as the Dice loss is better suited for class-imbalanced problems, but otherwise, keep Davari et al. (2021)'s loss as is. In order to

get the weights for the prediction, the front label is first dilated (see Eq. 6, where $\delta_w$ indicates the morphological dilation with a rectangular structuring element of size $w \times w$). Second, the Euclidean distance transform (EDT) is applied on the dilated label, which is then divided by a relaxation factor $R$ and fed into a sigmoid function, which is denoted by $\sigma$ (see Eq. 7). The final weight map is the sum of the output of the sigmoid function and the dilated front label, which is first inverted and then scaled by a factor $k$ (see Eq. 8). The improved distance map loss is visualized in Fig. 7. The hyperparameters $w$, $R$, and $k$ were set to

5, 1 and 0.1 respectively after empirical evaluations.

$$x_{\text{dil}} = \delta_w(x) \tag{6}$$

$$x_{\text{edt}} = \sigma \left( \frac{\text{EDT}(x_{\text{dil}})}{R} \right) \tag{7}$$

$$\text{weights} = x_{\text{edt}} + k \cdot (1 - x_{\text{dil}}) \tag{8}$$

Both models were trained five times for 150 epochs using early stopping with a patience of 30 epochs. For the zones model,

the early stopping criterion is the mean validation multi-class IoU, while the criterion is the mean validation loss for the front model. The weights of the best epoch of each training round are used to evaluate the models' performance on the test set. The best epoch is determined by the highest mean IoU on the validation set for zone segmentation and by the lowest validation loss for front segmentation. Finally, the mean and standard deviation of the two models' performances over the five training rounds are calculated.





$$\text{Loss} = \text{Dice}(\quad \circ \quad , \quad )$$

**Figure 7.** Visualization of the improved distance map loss. The left image shows the weights, the middle is the network's prediction, and on the right is the front label. ∘ denotes the Hadamard product.

## 5.3 Results


In this section, we evaluate the performance of our vanilla models, present the results quantitatively and qualitatively, and discuss these results objectively. First and foremost, we present our results on the test set. As our test set is out-of-sample, i. e., the study sites in the test set were not covered in the training set, the test set is entirely independent of the train set and hence, qualifies for estimating the generalizability of our trained models to new unseen glaciers. Some previous calving front

delineation studies (Cheng et al., 2021; Davari et al., 2022, 2021; Hartmann et al., 2021; Holzmann et al., 2021; Marochov et al., 2021; Periyasamy et al., 2022; Zhang et al., 2019) used in-sample test sets, i. e., the test set includes images from the same glaciers as covered in the train set but from different time points. Deep learning models will produce more precise front delineations on in-sample test sets compared to out-of-sample test sets (Marochov et al., 2021), as the generalization gap between train and test set is smaller for in-sample test sets (Quinonero-Candela et al., 2008). If a model is meant only to

delineate fronts from glaciers covered in the train set, it is legitimate to evaluate the model on an in-sample test set. However, if the model shall also be applied to new glaciers, it is crucial to evaluate it on an out-of-sample test set. To approximate an evaluation on an in-sample test set, we also present our results on the validation set (see Sect. A). Bear in mind that we used this validation set to optimize our hyper-parameters. Hence, the results on the validation set do not accurately reflect the performance of our model on in-sample test data but likely lead to somewhat biased results.

### 5.3.1 Zone segmentation

We begin our evaluation with the segmentation results of the zones model. Therefore, we calculate the segmentation metrics presented in Sect. 5.1.1. The resulting values for the test set are given in Table 5. All metrics show the highest values for the NA area, which was to be expected, as areas outside the swath, lay-over regions, and SAR shadows are relatively easy to distinguish in the images compared to the other classes. Overall, the precision is higher than the recall except for the glacier

class. The rock outcrop class has the highest drop between precision and recall implying that a considerable fraction of the predicted rock pixels is actually rock, but only a smaller fraction of the real rock outcrop was predicted to be rock. The rock outcrop class also has the lowest F1 score and lowest IoU. The evaluation on the validation set results in a precision of $94.5 \pm$ 0.2, a recall of $91.8 \pm$, an F1 score of $92.9 \pm 0.3$ and an IoU of $88.0 \pm 0.3$. The complete evaluation on the validation set is given in Table A1. All metrics are considerably higher for the validation set than for the out-of-sample test set.





| Scope | Precision ↑ | Recall ↑ | F1 Score ↑ | IoU ↑ |
|---|---|---|---|---|
| NA Area | $99.5 \pm 0.1$ | $91.2 \pm 1.3$ | $94.8 \pm 0.8$ | $90.9 \pm 1.3$ |
| Rock Outcrop | $82.0 \pm 0.5$ | $59.6 \pm 1.3$ | $67.9 \pm 0.8$ | $53.5 \pm 0.7$ |
| Glacier | $74.5 \pm 0.7$ | $89.5 \pm 1.1$ | $80.9 \pm 0.3$ | $68.5 \pm 0.5$ |
| Ocean and Ice Melange | $80.9 \pm 2.2$ | $78.3 \pm 3.1$ | $76.8 \pm 1.6$ | $66.0 \pm 1.5$ |
| All Zones | $84.2 \pm 0.5$ | $79.6 \pm 0.9$ | $80.1 \pm 0.5$ | $69.7 \pm 0.6$ |

**Table 5.** Segmentation results on the test set for the zones model in percent. The scope implicates the multi-class or respective class-wise metric.

| Precision ↑ | Recall ↑ | F1 Score ↑ | IoU ↑ |
|---|---|---|---|
| $2.4 \pm 0.3$ | $57.3 \pm 5.4$ | $4.3 \pm 0.5$ | $2.2 \pm 0.3$ |

**Table 6.** Segmentation results on the test set for the front model in percent.

### 5.3.2 Front segmentation

For the evaluation of the front segmentation model, we do not distinguish between multi-class and class-wise metrics, as the front segmentation is a binary task. Precision, recall, F1 score and IoU on the test set are given in Table 6. The recall is the highest metric with $57.3 \pm 5.4$, signifying that more than half of the ground truth front pixels are covered by the predicted front. However, all values are considerably lower than for the zone segmentation. The low values are explained by the composition and structure of the label, as the front label shows a high class imbalance, i.e., the background occupies significantly more pixels than the front, which is just a one-pixel-wide line. The one-pixel-wide structure of the front also leads to problems with the segmentation metrics. If, for example, the predicted front lies right next to the ground truth front separated by only one pixel, the IoU will be zero, as it measures the fraction of overlap between prediction and ground truth. Therefore, even though the prediction would be very close to the ground truth, the IoU would not reflect the quality of this prediction. Hence, these results should not be compared directly to the zone segmentation metrics but only to other front segmentation results. Instead, the proposed MDE given in Sect. 5.1.2 should be preferred for comparisons between the different segmentation models. The front segmentation metrics on the validation set are shown in Table A2.

### 5.3.3 Front delineation

After post-processing, the MDE can be calculated. Table 7 gives the MDEs for zone and front models and additionally breaks down the metric by glacier and season. Overall, the zone segmentation model leads to front predictions that are, on average,





| Network | | MDE ↓ | ∅ ↓ | Summer MDE ↓ | Summer ∅ ↓ | Winter MDE ↓ | Winter ∅ ↓ |
|---|---|---|---|---|---|---|---|
| Front | All | $887 \pm 189$ | $7 \pm 3 \in 122$ | $738 \pm 111$ | $4 \pm 1 \in 68$ | $1,054 \pm 308$ | $4 \pm 2 \in 54$ |
| | Columbia | $1,032 \pm 227$ | $2 \pm 1 \in 65$ | $907 \pm 131$ | $0 \pm 0 \in 28$ | $1,157 \pm 350$ | $2 \pm 1 \in 37$ |
| | Mapple | $150 \pm 24$ | $6 \pm 2 \in 57$ | $140 \pm 26$ | $2 \pm 1 \in 40$ | $173 \pm 33$ | $2 \pm 1 \in 17$ |
| Zones | All | $753 \pm 76$ | $1 \pm 1 \in 122$ | $732 \pm 93$ | $1 \pm 1 \in 68$ | $776 \pm 65$ | $0 \pm 0 \in 54$ |
| | Columbia | $840 \pm 84$ | $0 \pm 0 \in 65$ | $854 \pm 111$ | $0 \pm 0 \in 28$ | $826 \pm 66$ | $0 \pm 0 \in 37$ |
| | Mapple | $287 \pm 48$ | $0 \pm 1 \in 57$ | $262 \pm 29$ | $0 \pm 1 \in 40$ | $340 \pm 93$ | $0 \pm 0 \in 17$ |

**Table 7.** Mean distance errors (MDEs) on the test set in meters, also differentiated by glacier and season. ∅ stands for the number of images for which no front was predicted. The number after ∈ denotes how many images of the category (given glacier and season) were present in the test set.

by 143 meters closer to the front than the front segmentation model. Interestingly, however, the latter yields more accurate predictions for the Mapple Glacier than the zone segmentation model. Qualitative results of the front models are shown in Fig. 8. Figure 8 (a) and (b) are examples of accurate front delineations. Figure 8 (c) shows an inaccurate front delineation. Firstly, part of the coastline is confused as calving front, and secondly, only one of the three calving fronts of the Columbia Glacier is
recognized. The two unidentified calving fronts contribute significantly to the MDE. As the MDE is symmetric, for each pixel in the two unidentified ground truth fronts, the Euclidean distance to the closest pixel in the predicted front contributes to the MDE, and, obviously, the closest pixels are far away from the closest identified calving front pixels. Figure 8 (d) gives an example where sea ice was confused with a calving front. In general, predictions for images from summer are more accurate than images from winter, as can be seen in Table 7. This seasonal gap can be explained by sea ice forming in front of the
calving front during winter, with similar back-scattering properties as the glacier. Therefore, sea ice makes an accurate calving front delineation more difficult. Qualitative results of the zone segmentation model are given in Fig. 9. Figure 9 (a) and (b) are examples of accurate front delineations, whereas Fig. 9 (c) and (d) show confusions with sea ice again.

Table 8 breaks the MDEs down by glacier and satellite, showing that predictions for Sentinel-1 images are by far the least accurate over the complete test set. The reason for the high MDEs for Sentinel-1 is that in most images at Columbia Glacier,
only one of the three calving fronts is identified, and as explained earlier, the other two fronts negatively impact the MDE. Front model predictions for Sentinel-1 images of Mapple Glacier only have a marginally higher MDE than images from other satellites, while zone model predictions for Sentinel-1 images of Mapple Glacier show the lowest MDE across all satellites. In summary, this shows that the sensor influences the quality of the front delineation but that the glacier geometry has a greater impact.
A breakdown of the MDEs by glaciers and sensor spatial resolution is shown in Table 9. A spatial resolution of $20\,\mathrm{m}^2$ is associated with the highest MDEs. When the images with this resolution are split up into glaciers, it becomes apparent that



Earth System
Science
Data

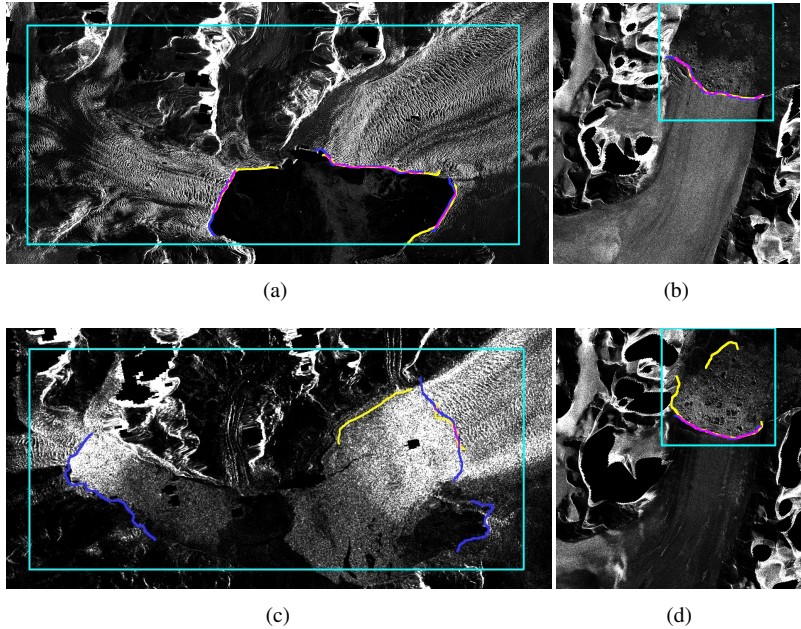

(a)    (b)

(c)    (d)

**Figure 8.** Visualization of the front segmentation models' performance on the test set. Blue represents the ground truth, yellow the prediction, and pink the overlap of ground truth and prediction. The turquoise rectangle is the bounding box explained in Sect. 4.3. (a) is an image of the Columbia Glacier acquired on the 19th of March 2012 by the TDX satellite. (b) is an image of the Mapple Glacier acquired on the 2nd of November 2009 by the TSX satellite and (c) is an image of the Columbia Glacier acquired on the 6th of January 2018 by the Sentinel-1 satellite. (d) is an image of the Mapple Glacier acquired on the 30th of June 2013 by the TSX satellite. The front prediction and ground truth of (a), (b), and (d) are dilated with a $9 \times 9$ kernel and (c) with a $3 \times 3$ kernel to enhance the visibility of the region of interest. The images of the Columbia Glacier are cropped to the region of interest for visualization purposes.

the high MDEs result from images of the Columbia Glacier. The lowest average MDEs are observed for a spatial resolution of $17\,\text{m}^2$. This can be explained by the fact that the test set for a resolution of $17\,\text{m}^2$ only includes images of the Mapple Glacier. Hence, we can conclude that the glacier geometry has a higher impact on the front delineation performance than the spatial resolution of the image.

The predictions on the validation set receive an average MDE of $391\,\text{m} \pm 94\,\text{m}$ for the front model and $449\,\text{m} \pm 31\,\text{m}$ for the zone model, which is considerably lower than on the test set. The visualizations, Figures A1 and A2, show that the front delineation is very accurate, but part of the coastlines are predicted as calving front. Hence, with a post-processing scheme that can eliminate calving front predictions on the coastline, the MDE would even be smaller. The complete evaluations on the validation set are given in Tables A3, A4, and A5.



| Network | Glacier | | Sentinel-1 | ENVISAT | ERS-1/2 | PALSAR | TSX/TDX |
|---|---|---|---|---|---|---|---|
| Front | All | MDE ↓ | 2,806 ± 300 | 191 ± 32 | 127 ± 38 | 197 ± 41 | 663 ± 188 |
| | | ∅ ↓ | 2 ± 1 ∈ 33 | 2 ± 2 ∈ 10 | 0 ± 0 ∈ 2 | 3 ± 2 ∈ 8 | 0 ± 0 ∈ 69 |
| | Columbia | MDE ↓ | 3,537 ± 422 | / | / | / | 744 ± 218 |
| | | ∅ ↓ | 2 ± 1 ∈ 18 | / | / | / | 0 ± 0 ∈ 47 |
| | Mapple | MDE ↓ | 206 ± 33 | 191 ± 32 | 127 ± 38 | 197 ± 41 | 129 ± 33 |
| | | ∅ ↓ | 0 ± 0 ∈ 15 | 2 ± 2 ∈ 10 | 0 ± 0 ∈ 2 | 3 ± 2 ∈ 8 | 0 ± 0 ∈ 22 |
| Zones | All | MDE ↓ | 2,201 ± 246 | 493 ± 119 | 404 ± 172 | 437 ± 42 | 547 ± 61 |
| | | ∅ ↓ | 0 ± 0 ∈ 33 | 0 ± 0 ∈ 10 | 0 ± 0 ∈ 2 | 0 ± 0 ∈ 8 | 0 ± 0 ∈ 69 |
| | Columbia | MDE ↓ | 2,587 ± 299 | / | / | / | 587 ± 67 |
| | | ∅ ↓ | 0 ± 0 ∈ 18 | / | / | / | 0 ± 0 ∈ 47 |
| | Mapple | MDE ↓ | 141 ± 29 | 493 ± 119 | 404 ± 172 | 437 ± 42 | 246 ± 57 |
| | | ∅ ↓ | 0 ± 0 ∈ 15 | 0 ± 0 ∈ 10 | 0 ± 0 ∈ 2 | 0 ± 0 ∈ 8 | 0 ± 0 ∈ 22 |

**Table 8.** Mean distance errors (MDEs) on the test set in meters, differentiated by glacier and satellite. ∅ stands for the number of images for which no front was predicted. The number after ∈ denotes how many images of the category (given glacier and satellite) were present in the test set.

| Network | | 20 | | 17 | | 7 | |
|---|---|---|---|---|---|---|---|
| | | MDE ↓ | ∅ ↓ | MDE ↓ | ∅ ↓ | MDE ↓ | ∅ ↓ |
| Front | All | 2,436 ± 289 | 4 ± 2 ∈ 45 | 197 ± 41 | 3 ± 2 ∈ 8 | 663 ± 188 | 0 ± 0 ∈ 69 |
| | Columbia | 3,537 ± 422 | 2 ± 1 ∈ 18 | / | / | 744 ± 218 | 0 ± 0 ∈ 47 |
| | Mapple | 192 ± 26 | 2 ± 2 ∈ 27 | 197 ± 41 | 3 ± 2 ∈ 8 | 129 ± 33 | 0 ± 0 ∈ 22 |
| Zones | All | 1,939 ± 220 | 0 ± 0 ∈ 45 | 437 ± 42 | 0 ± 0 ∈ 8 | 547 ± 61 | 0 ± 0 ∈ 69 |
| | Columbia | 2,587 ± 299 | 0 ± 0 ∈ 18 | / | / | 587 ± 67 | 0 ± 0 ∈ 47 |
| | Mapple | 323 ± 69 | 0 ± 0 ∈ 27 | 437 ± 42 | 0 ± 0 ∈ 8 | 246 ± 57 | 0 ± 0 ∈ 22 |

**Table 9.** Mean distance errors (MDEs) on the test set in meters, differentiated by glacier and resolution. ∅ stands for the number of images for which no front was predicted. The number after ∈ denotes how many images of the category (given glacier and resolution) were present in the test set.

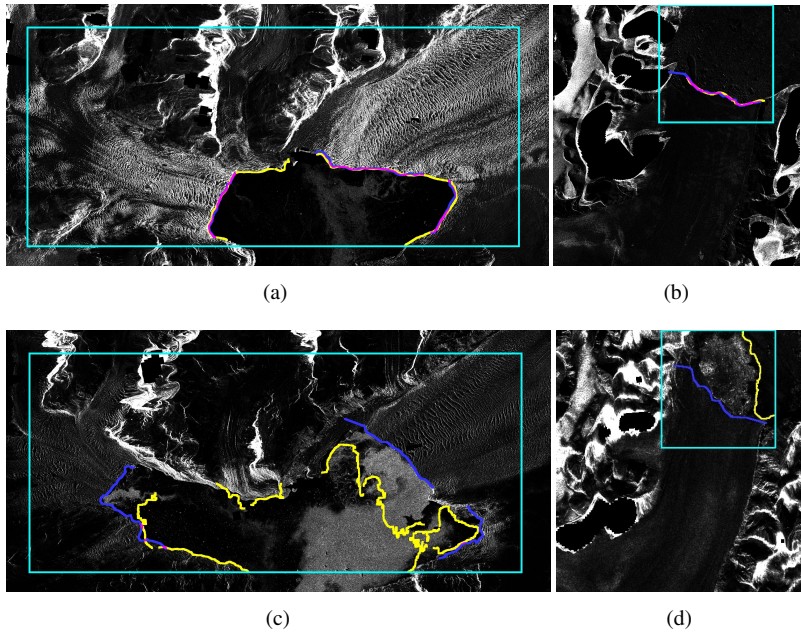

Figure 9. Visualization of the zone segmentation models' performance on the test set. Blue represents the ground truth, yellow the prediction, and pink the overlap of ground truth and prediction. The turquoise rectangle is the bounding box explained in Sect. 4.3. (a) is an image of the Columbia Glacier acquired on the 19th of March 2012 by the TDX satellite. (b) is an image of the Mapple Glacier acquired on the fifth of November 2010 by the TSX satellite. (c) is an image of the Columbia Glacier acquired on the 17th of April 2016 by the TDX satellite. (d) is an image of the Mapple Glacier acquired on the 21st of November 2007 by the PALSAR satellite. The front prediction and ground truth of (a), (b), and (c) are dilated with a $9 \times 9$ kernel and (d) with a $3 \times 3$ kernel to enhance the visibility of the region of interest. The images of the Columbia Glacier are cropped to the region of interest for visualization purposes.

## 5.4 Discussion and outlook

In Sect. 5.3, major performance differences between results on different satellites, resolutions, and glaciers became apparent, which can mostly be attributed to the complexity gap between the Mapple and Columbia glaciers in the test set. The Columbia Glacier consists of three separate calving fronts in contrast to Mapple, which only features one calving front. In addition, 470 the Columbia Glacier's calving fronts exhibit greater variability in their structure and curvature than Mapple's calving front. We, therefore, conclude that the complexity of the glacier, including the number and shape of calving fronts, has a significant impact on the delineation performance of a deep learning model.

Performance differences between zone and front models are also remarkable. The zone models' predictions lie mainly close to the calving front or in the ocean and only rarely far away on the other side of the image between, e. g., rock outcrop and 475 NA area or rock outcrop and glacier. On the other hand, the front models' predictions are not that constraint in the region, but if they hit the correct calving front region, the prediction lies more accurately on the true calving front than the zone models'



prediction. Therefore, a multi-task learning approach could be beneficial for future work as the two tasks, zone segmentation and front segmentation, excel in different aspects and can learn from each other here.

From the class-wise segmentation metrics presented in Sect. 5.3.1, it is notable that the rock outcrop class was the least
accurately predicted. One reason is that the rock outcrop is static for each glacier, i. e., no adjustments were made for the individual image labels even if part or the whole rock outcrop was covered by snow or new rocks became visible as the ice receded throughout time. Hence, this drop in performance comes from the labels themselves. We nevertheless decided to use static rock outcrops as an accurate rock outcrop prediction is not the aim of our work, but only contributes to an accurate front delineation, and manually labelling the rock as done for the calving front would have been beyond the scope of this paper.

Two weaknesses that can be addressed in the future include low performance on images with sea ice and confusion of the coastline with the calving front. Weak performance on images with sea ice might be compensated for by augmenting winter images more frequently than summer images or giving higher weight to the loss of winter images. The confusion of the coastline with the calving front could be remedied by improved post-processing such as masking with a rock outcrop, which of course, would have to be available for each new glacier.

All in all, the vanilla models presented in this paper provide a starting point for accurate segmentation of calving fronts, and the introduced dataset enables training, testing, and comparison of deep learning calving front delineation models.

## 6 Conclusions

In this paper, we introduce a benchmark dataset for calving front delineation of marine-terminating glaciers using SAR imagery. It is the first to provide long-term calving front information from multi-mission data. Furthermore, besides the provided
segmentation labels, the dataset contains the corresponding preprocessed and geo-referenced SAR images as PNG files. The dataset's ease of access allows scientists to focus on the deep learning model and invites researchers from related scientific fields such as data science to contribute to this urgent task. The calving front position is an indicator of frontal ablation, which is a key parameter of the total glacier mass balance. With this dataset, we aim to enable the training of deep-learning models capable of accurately delineating calving fronts of glaciers around the world at any time, permitting a spatio-temporal quan-
tification of calving rates. The dataset includes SAR imagery of glaciers from Antarctica, Greenland and Alaska. With its light in-dependency and cloud penetration ability, SAR imagery permits continuous front tracking across all seasons. The test set includes the Mapple Glacier located in Antarctica and the Columbia Glacier in Alaska. Glaciers from different regions in the test set guarantee that, during testing, the wide applicability of the model is verified. Moreover, no images from these two glaciers are included in the train set, making it an out-of-sample test set. Hence, the models' generalizability to unseen glaciers
and regions also outside Antarctica and Greenland can be ensured during testing. Additionally, with the Columbia Glacier, the test set includes challenging samples, as Columbia Glacier has three calving fronts with highly variable geometry. Therefore, a low MDE on these samples indicates the robustness of the trained model to variations in the shape and number of calving fronts in the given images. The two labels in the dataset, zone and front segmentation, supply comprehensive information on the calving front delineation task and allow to approach the task in different ways. The well-documented and easy-to-use code





of the two vanilla models introduced in this paper ensures the reproducibility of all experiments and provides a starting point for an accurate front delineation method that can easily be extended. The performance of different models can be quantified and compared to other models consistently with the presented evaluation metrics. The resulting MDE of $150\,\mathrm{m} \pm 24\,\mathrm{m}$ for the Mapple Glacier and $840\,\mathrm{m} \pm 84\,\mathrm{m}$ for the Columbia Glacier build a sound baseline for future calving front delineation models.

*Code and data availability.* The code is publicly available on GitHub under https://github.com/Nora-Go/Calving_Fronts_and_Where_to_
Find_Them. The exact version of the model used to produce the results presented in this paper is archived on Zenodo (Gourmelon et al., 2022a). The dataset is publicly available at PANGAEA (Gourmelon et al., 2022b) under https://doi.pangaea.de/10.1594/PANGAEA.940950 (DOI will be registered upon acceptance of the paper).

## Appendix A: Validation results

This section depicts the vanilla models' results on the in-sample validation set. The validation set was used to optimize the
models' hyper-parameters, and, hence, the results are biased.

## A1  Zone segmentation

Table A1 presents the evaluation of the zone models' segmentation results on the validation set. The segmentation metrics introduced in Sect. 5.1.1 are employed for the evaluation.

| Scope | Precision ↑ | Recall ↑ | F1 Score ↑ | IoU ↑ |
|---|---|---|---|---|
| All Zones | 94.5 ± 0.2 | 91.8 ± 0.5 | 92.9 ± 0.3 | 88.0 ± 0.3 |
| NA Area | 96.0 ± 1.0 | 91.2 ± 2.3 | 92.9 ± 1.3 | 89.3 ± 1.4 |
| Rock Outcrop | 89.4 ± 0.8 | 82.6 ± 1.1 | 85.7 ± 0.3 | 75.8 ± 0.4 |
| Glacier | 95.6 ± 0.1 | 97.5 ± 0.2 | 96.5 ± 0.1 | 93.3 ± 0.1 |
| Ocean and Ice Melange | 97.3 ± 0.3 | 96.0 ± 0.4 | 96.5 ± 0.2 | 93.5 ± 0.3 |

**Table A1.** Segmentation results on the validation set for the zones model in percent. The scope implicates the multi-class or respective class-wise metric.

## A2  Front segmentation

Table A2 gives the precision, recall, F1 score and IoU of the front models' segmentation results on the validation set.





| Precision ↑ | Recall ↑ | F1 Score ↑ | IoU ↑ |
|---|---|---|---|
| 2.3 ± 0.2 | 67.5 ± 1.4 | 4.5 ± 0.4 | 2.3 ± 0.2 |

**Table A2.** Segmentation results on the validation set for the front model in percent.

## A3 Front delineation

The MDEs on the post-processed validation set results are given in Table A3, which also differentiates between glaciers and seasons. Table A4 breaks these results down by glacier and satellite and Table A5 by glacier and resolution. Visualizations of the prediction results on the validation set can be seen in Figures A1 and A2.

| Network | | MDE ↓ | ∅ ↓ | Summer MDE ↓ | ∅ ↓ | Winter MDE ↓ | ∅ ↓ |
|---|---|---|---|---|---|---|---|
| Front | All | 391 ± 94 | 0 ± 0 ∈ 56 | 466 ± 89 | 0 ± 0 ∈ 36 | 315 ± 106 | 0 ± 0 ∈ 20 |
| | Crane | 593 ± 123 | 0 ± 0 ∈ 8 | 695 ± 128 | 0 ± 0 ∈ 6 | 368 ± 135 | 0 ± 0 ∈ 2 |
| | DBE | 561 ± 144 | 0 ± 0 ∈ 10 | 717 ± 185 | 0 ± 0 ∈ 6 | 225 ± 66 | 0 ± 0 ∈ 4 |
| | JAC | 222 ± 21 | 0 ± 0 ∈ 16 | 221 ± 16 | 0 ± 0 ∈ 6 | 222 ± 25 | 0 ± 0 ∈ 10 |
| | Jorum | 276 ± 99 | 0 ± 0 ∈ 8 | 276 ± 99 | 0 ± 0 ∈ 8 | / | / |
| | SI | 741 ± 383 | 0 ± 0 ∈ 14 | 730 ± 255 | 0 ± 0 ∈ 10 | 774 ± 652 | 0 ± 0 ∈ 4 |
| Zones | All | 449 ± 31 | 0 ± 0 ∈ 56 | 563 ± 43 | 0 ± 0 ∈ 36 | 318 ± 26 | 0 ± 0 ∈ 20 |
| | Crane | 784 ± 79 | 0 ± 0 ∈ 8 | 852 ± 106 | 0 ± 0 ∈ 6 | 580 ± 170 | 0 ± 0 ∈ 2 |
| | DBE | 1,043 ± 85 | 0 ± 0 ∈ 10 | 1,095 ± 121 | 0 ± 0 ∈ 6 | 907 ± 30 | 0 ± 0 ∈ 4 |
| | JAC | 150 ± 16 | 0 ± 0 ∈ 16 | 169 ± 25 | 0 ± 0 ∈ 6 | 138 ± 12 | 0 ± 0 ∈ 10 |
| | Jorum | 402 ± 73 | 0 ± 0 ∈ 8 | 402 ± 73 | 0 ± 0 ∈ 8 | / | / |
| | SI | 737 ± 73 | 0 ± 0 ∈ 14 | 777 ± 48 | 0 ± 0 ∈ 10 | 669 ± 124 | 0 ± 0 ∈ 4 |

**Table A3.** Mean distance errors (MDEs) on the validation set in meters, also differentiated by glacier and season. ∅ stands for the number of images for which no front was predicted. The number after ∈ denotes how many images of the category (given glacier and season) were present in the validation set.

*Author contributions.* **Nora Gourmelon**: Conceptualization, Methodology, Software, Project administration, Writing - Original draft preparation. **Thorsten Seehaus**: Data curation, Writing - Original draft preparation. **Matthias Braun**: Supervision, Writing – review & editing. **Andreas Maier**: Supervision, Writing – review & editing. **Vincent Christlein**: Supervision, Validation, Writing - Original draft preparation.



| Network | | | RADARSAT-1 | Sentinel-1 | ENVISAT | ERS-1/2 | PALSAR | TSX/TDX |
|---|---|---|---|---|---|---|---|---|
| **Front** | All | MDE ↓ | 545 ± 142 | 205 ± 25 | 539 ± 158 | 840 ± 134 | 1,427 ± 239 | 341 ± 89 |
| | | ∅ ↓ | 0 ± 0 ∈ 4 | 0 ± 0 ∈ 1 | 0 ± 0 ∈ 9 | 0 ± 0 ∈ 5 | 0 ± 0 ∈ 2 | 0 ± 0 ∈ 35 |
| | Crane | MDE ↓ | / | / | 1,147 ± 625 | 303 ± 81 | / | 569 ± 109 |
| | | ∅ ↓ | / | / | 0 ± 0 ∈ 1 | 0 ± 0 ∈ 1 | / | 0 ± 0 ∈ 6 |
| | DBE | MDE ↓ | 358 ± 145 | / | 366 ± 137 | / | 614 ± 168 | 663 ± 211 |
| | | ∅ ↓ | 0 ± 0 ∈ 1 | / | 0 ± 0 ∈ 4 | / | 0 ± 0 ∈ 1 | 0 ± 0 ∈ 4 |
| | JAC | MDE ↓ | / | / | / | / | / | 222 ± 21 |
| | | ∅ ↓ | / | / | / | / | / | 0 ± 0 ∈ 16 |
| | Jorum | MDE ↓ | / | 205 ± 25 | 442 ± 236 | / | / | 252 ± 91 |
| | | ∅ ↓ | / | 0 ± 0 ∈ 1 | 0 ± 0 ∈ 2 | / | / | 0 ± 0 ∈ 5 |
| | SI | MDE ↓ | 575 ± 153 | / | 600 ± 261 | 916 ± 143 | 1,780 ± 291 | 725 ± 590 |
| | | ∅ ↓ | 0 ± 0 ∈ 3 | / | 0 ± 0 ∈ 2 | 0 ± 0 ∈ 4 | 0 ± 0 ∈ 1 | 0 ± 0 ∈ 4 |
| **Zones** | All | MDE ↓ | 900 ± 118 | 164 ± 45 | 776 ± 136 | 846 ± 157 | 1229 ± 215 | 365 ± 26 |
| | | ∅ ↓ | 0 ± 0 ∈ 4 | 0 ± 0 ∈ 1 | 0 ± 0 ∈ 9 | 0 ± 0 ∈ 5 | 0 ± 0 ∈ 2 | 0 ± 0 ∈ 35 |
| | Crane | MDE ↓ | / | / | 393 ± 356 | 154 ± 27 | / | 837 ± 73 |
| | | ∅ ↓ | / | / | 0 ± 0 ∈ 1 | 0 ± 0 ∈ 1 | / | 0 ± 0 ∈ 6 |
| | DBE | MDE ↓ | 812 ± 78 | / | 938 ± 140 | / | 1,434 ± 164 | 1,065 ± 126 |
| | | ∅ ↓ | 0 ± 0 ∈ 1 | / | 0 ± 0 ∈ 4 | / | 0 ± 0 ∈ 1 | 0 ± 0 ∈ 4 |
| | JAC | MDE ↓ | / | / | / | / | / | 150 ± 16 |
| | | ∅ ↓ | / | / | / | / | / | 0 ± 0 ∈ 16 |
| | Jorum | MDE ↓ | / | 164 ± 45 | 155 ± 55 | / | / | 441 ± 80 |
| | | ∅ ↓ | / | 0 ± 0 ∈ 1 | 0 ± 0 ∈ 2 | / | / | 0 ± 0 ∈ 5 |
| | SI | MDE ↓ | 912 ± 135 | / | 801 ± 209 | 939 ± 186 | 1,118 ± 331 | 605 ± 82 |
| | | ∅ ↓ | 0 ± 0 ∈ 3 | / | 0 ± 0 ∈ 2 | 0 ± 0 ∈ 4 | 0 ± 0 ∈ 1 | 0 ± 0 ∈ 4 |

**Table A4.** Mean distance errors (MDEs) on the validation set in meters, differentiated by glacier and satellite. ∅ stands for the number of images for which no front was predicted. The number after ∈ denotes how many images of the category (given glacier and satellite) were present in the validation set.





| Network | Glacier | | 20 | 17 | 12 | 7 | 6 |
|---|---|---|---|---|---|---|---|
| Front | All | MDE ↓ | 636 ± 130 | 1,427 ± 239 | 545 ± 142 | 605 ± 264 | 222 ± 21 |
| | | ∅ ↓ | 0 ± 0 ∈ 15 | 0 ± 0 ∈ 2 | 0 ± 0 ∈ 4 | 0 ± 0 ∈ 19 | 0 ± 0 ∈ 16 |
| | Crane | MDE ↓ | 754 ± 337 | / | / | 569 ± 109 | / |
| | | ∅ ↓ | 0 ± 0 ∈ 2 | / | / | 0 ± 0 ∈ 6 | / |
| | DBE | MDE ↓ | 366 ± 137 | 614 ± 168 | 358 ± 145 | 663 ± 211 | / |
| | | ∅ ↓ | 0 ± 0 ∈ 4 | 0 ± 0 ∈ 1 | 0 ± 0 ∈ 1 | 0 ± 0 ∈ 4 | / |
| | JAC | MDE ↓ | / | / | / | / | 222 ± 21 |
| | | ∅ ↓ | / | / | / | / | 0 ± 0 ∈ 16 |
| | Jorum | MDE ↓ | 369 ± 165 | / | / | 252 ± 91 | / |
| | | ∅ ↓ | 0 ± 0 ∈ 3 | / | / | 0 ± 0 ∈ 5 | / |
| | SI | MDE ↓ | 774 ± 185 | 1,780 ± 291 | 575 ± 153 | 725 ± 590 | / |
| | | ∅ ↓ | 0 ± 0 ∈ 6 | 0 ± 0 ∈ 1 | 0 ± 0 ∈ 3 | 0 ± 0 ∈ 4 | / |
| Zones | All | MDE ↓ | 782 ± 134 | 1,229 ± 215 | 900 ± 118 | 718 ± 62 | 718 ± 62 |
| | | ∅ ↓ | 0 ± 0 ∈ 15 | 0 ± 0 ∈ 2 | 0 ± 0 ∈ 4 | 0 ± 0 ∈ 19 | 0 ± 0 ∈ 16 |
| | Crane | MDE ↓ | 270 ± 179 | / | / | 837 ± 73 | / |
| | | ∅ ↓ | 0 ± 0 ∈ 2 | / | / | 0 ± 0 ∈ 6 | / |
| | DBE | MDE ↓ | 938 ± 140 | 1,434 ± 164 | 812 ± 78 | 1,065 ± 126 | / |
| | | ∅ ↓ | 0 ± 0 ∈ 4 | 0 ± 0 ∈ 1 | 0 ± 0 ∈ 1 | 0 ± 0 ∈ 4 | / |
| | JAC | MDE ↓ | / | / | / | / | 150 ± 16 |
| | | ∅ ↓ | / | / | / | / | 0 ± 0 ∈ 16 |
| | Jorum | MDE ↓ | 158 ± 46 | / | / | 441 ± 80 | / |
| | | ∅ ↓ | 0 ± 0 ∈ 3 | / | / | 0 ± 0 ∈ 5 | / |
| | SI | MDE ↓ | 874 ± 170 | 1,118 ± 331 | 912 ± 135 | 605 ± 82 | / |
| | | ∅ ↓ | 0 ± 0 ∈ 6 | 0 ± 0 ∈ 1 | 0 ± 0 ∈ 3 | 0 ± 0 ∈ 4 | / |

**Table A5.** Mean distance errors (MDEs) on the validation set in meters, differentiated by glacier and resolution. ∅ stands for the number of images for which no front was predicted. The number after ∈ denotes how many images of the category (given glacier and resolution) were present in the validation set.



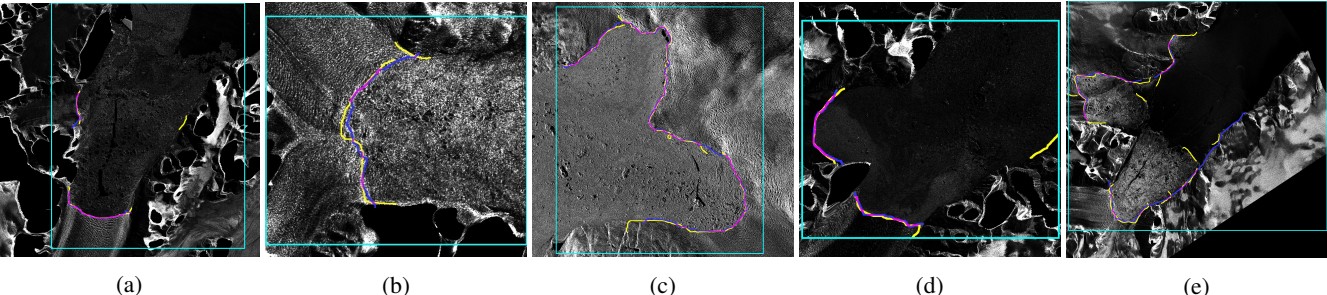

(a)      (b)      (c)      (d)      (e)

**Figure A1.** Visualization of the front segmentation models' performance on the validation set. Blue represents the ground truth, yellow the prediction, and pink the overlap of ground truth and prediction. The turquoise rectangle is the bounding box explained in Sect. 4.3. (a) is an image of the Crane Glacier acquired on the eighth of December 2010 by the TSX satellite. (b) is an image of the DBE Glacier acquired on the 21st of July 2004 by the ENVISAT satellite. (c) is an image of the Jakobshavn glacier acquired on the third of October 2012 by the TSX satellite. (d) is an image of the Jorum Glacier acquired on the 19th of December 2010 by the TSX satellite. (e) is an image of the SI Glacier acquired on the 19th of October 2013 by the TSX satellite. The images are cropped to the region of interest for visualization purposes. The front prediction and ground truth of (a), (c), (d), and (e) are dilated with a $9 \times 9$ kernel and (b) with a $3 \times 3$ kernel to enhance the visibility.

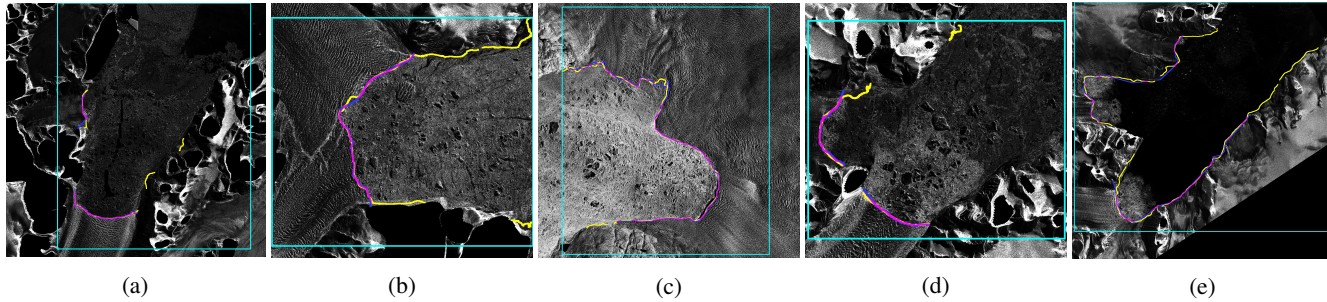

(a)      (b)      (c)      (d)      (e)

**Figure A2.** Visualization of the zone segmentation models' performance on the validation set. Blue represents the ground truth, yellow the prediction, and pink the overlap of ground truth and prediction. The turquoise rectangle is the bounding box explained in Sect. 4.3. (a) is an image of the Crane Glacier acquired on the eighth of December 2010 by the TSX satellite. (b) is an image of the DBE Glacier acquired on the eleventh of July 2013 by the TSX satellite. (c) is an image of the Jakobshavn glacier acquired on the 15th of August 2009 by the TSX satellite. (d) is an image of the Jorum Glacier acquired on the first of September 2012 by the TSX satellite. (e) is an image of the SI Glacier acquired on the 19th of August 2011 by the TSX satellite. The images are cropped to the region of interest for visualization purposes. The front predictions and ground truths are dilated with a $9 \times 9$ kernel to enhance the visibility.





*Competing interests.* The authors declare that they have no conflict of interest.

*Acknowledgements.* We thank the Friedrich-Alexander-Universität for the funding under the Emerging Fields Initiative "Tapping the Poten-
tial of Earth Observations" and STAEDLER Foundation. We acknowledge the free provision of the SAR data via various proposals from
ESA, ASF and DLR.



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
