# Peer review of "Calving Fronts and Where to Find Them: A Benchmark Dataset and Methodology for Automatic Glacier Calving Front Extraction from SAR Imagery"

_Earth System Science Data, 2022_

## Author Comment (AC1)

We thank all reviewers for their thorough reviews. We carefully modified the paper according to their suggestions. Below, we provide a detailed list of our responses and the changes undertaken to the manuscript.

**Reviewer 1 - Major Concerns:**

**1.1** *Compared with previous studies, this manuscript has room to improve regarding innovation. The previous study has used deep learning to extract the glacier termini from multiple remote sensing datasets. If combining multi-sensor and long-term images brings innovations, the author needs to justify the reason (where are the challenges and how do you solve them).*

▸ Thank you for raising this point. This study does not focus on innovative deep learning techniques. Instead, the focus of this manuscript is to present a novel dataset that can be used to train and evaluate deep learning techniques. Training and testing on a common dataset ensures comparability between studies. Different models trained and evaluated on different datasets will get different error rates. It is, however, not clear if this performance difference results from the models or the dataset. It may well be that one of the used datasets is not representative, or that the test dataset is simpler (e. g., glaciers such as Mapple, where there is only one calving front with little curvature and variability). Therefore, a challenging benchmark data set is highly needed. We now emphasize this aim more clearly in change 1, change 2, deletion 1, change 3, addition 1, change 4, change 5, addition 4, addition 9, addition 10, and addition 11. Moreover, time series from multiple missions introduce new challenges for calving front segmentation, such as different spatial resolutions, different penetration depths or sensitivity to surface changes, different signal-to-noise ratios, and different geometries, topographic effects, shading, and overlay effects. We inserted this statement in addition 2.

**1.2** *The second concern is about the data quality and quantity. The uncertainty is high for one of the two test glaciers, and the terminus results are not usable for further analysis. Also, deep learning aims to produce a large number of termini by automating terminus delineation. But the number of terminus traces derived from this study is limited. Considering the ESSD is a journal focusing on data, I think the manuscript needs to improve in this regard.*

▸ The reviewer raises an important point. The dataset presented is not intended to be used directly for further analysis. Rather, it is intended to provide a basis for training and evaluating new deep-learning pipelines that can then be used to delineate calving fronts in new satellite imagery. As the reviewer correctly notes, the baseline model predictions for Columbia Glacier are not accurate enough to be used for further analysis. We make no claim that this is the case. With the base models, we aim to provide a foundation for future models and show how the dataset can be used in deep learning pipelines. We want to encourage further research to improve upon challenging samples (such as the Columbia Glacier) in the test set, as this is the only way to evaluate and thus ensure the generalizability of future models to difficult glaciers.

The quality of our annotated dataset has been verified by two additional persons. If there exist concerns about the annotations, please indicate the specific annotations that need to be revised. Concerning the quantity of data, a total of 681 annotated images with a mean size of $2182 \times 2010$ is a reasonable size for a dataset designated for training deep learning segmentation techniques. For comparison, the EuroSat benchmark dataset (https://github.com/phelber/eurosat) has 27 000 images of size $64 \times 64$. If we would divide our dataset into patches of $64 \times 64$, this would result in approximately 728 670 patches of this size making our dataset 27 times large than the EuroSat benchmark. Therefore, we think that the manuscript is a good fit for the special issue "Benchmark datasets and machine learning algorithms for Earth system science data". To enhance the comprehensibility of the manuscript, we highlight our intentions more clearly in change 1, change 2, deletion 1, change 3, addition 1, change 4, change 5, addition 4, addition 9, addition 10, and addition 11.

**Reviewer 1 - Specific Comments:**

**2.1** *Page 2 Line 58: It would be nice if the author could explain more about why the dataset from this study is beneficial for bridging the gap regarding the evaluation among different studies.*

▸ Thank you for this comment. With this dataset, different approaches for the detection of glacier calving fronts can be implemented, tested, and their performance fairly compared so that the most effective approach can be determined. Models trained and tested on different datasets are not comparable without re-training and testing on a common dataset. To clarify this, we added some explanations. Please see addition 3, change 1, and addition 9.

**2.2** *Page 10 Line 212: Please explain more about re-mapping.*

▸ Thank you for your request. By re-mapping, we just meant that we manually redrew all the front lines because we couldn't make the ones from Zhang et al. (2019) fit. We have clarified this in addition 5.

**2.3** *Page 10 Line 222: What is the rationale behind identifying four zones since this study will only pick the boundary between glacier and ocean.*

▸ Thank you for this question. There are two reasons why we decided to designate four zones instead of just two. First, if we were to divide the scene into ocean and non-ocean, the boundary between the two zones would be the entire coastline, not just the calving front. Therefore, it is important to include the rock outcrop as one zone. Second, SAR shadows, overlay regions, and areas outside the swath are shown as zero values in the satellite imagery. Labelling the same pattern (patches of zero values) once as 'ocean' and once as 'non-ocean' can confuse neural networks and affect their learning ability. In the early stages of creating the dataset, we saw evidence that there was indeed an advantage to including multiple zones instead of just two. To clarify this in the manuscript, we included addition 6.

**2.4** *Page 11 Line 246: Morphological dilation can resolve the imbalance to some degree but will also cause uncertainty in terminus delineation. The more balanced between positive and negative pixels, the larger uncertainties the terminus will have.*

▸ This raises an interesting point. We adopted this technique to cope with the class imbalance from Davari et al. (2022, 2021) and Periyasamy et al. (2022). Our baselines' mean distance error rates are significantly higher than $\frac{\text{structuring element size} - 1}{2} \cdot \max(\text{pixel resolution}) = 2 \cdot 20 = 40$. Therefore, we hypothesize that the dilation is not the main factor for the deviations to the ground truth front. Still, this is a great suggestion to improve over the baseline models. We will take it into account in our future work.

**2.5** *Page 13 Line 277: I like the idea of applying Gaussian importance weighting before taking the average. It would be nice if the author could provide more details.*

▸ Thank you! Gaussian importance weighting is done by element-wise multiplying each patch prediction with a Gaussian kernel of similar size. During patch merging, the sum of the respective pixel values is normalized by the sum of the corresponding Gaussian kernel's elements. We added this information in addition 7.

**2.6** *Page 15 Line 298: Is this a universal threshold? Please explain more about setting this threshold and what will happen if the threshold is too large. Usually, the threshold would be 0.5. The threshold can indeed be different as long as it is justified.*

▸ Thank you for pointing this out. The threshold was treated as a hyperparameter that we optimized on the validation set. We added this explanation to the manuscript in addition 8.

**2.7** *Page 21 Line 436: the uncertainty value might be wrong (150 m?). I couldn't find this value in Table 7.*

▸ We apologize, there was a mistake in the numbers! It should be 134 meters instead of 143 meters. The 134 meters are the difference between the front network's MDE over the complete test set and the zone network's MDE over the complete test set (887 – 753 = 134). We corrected this in the manuscript (see change 8) and thank the reviewer for pointing this out!

**2.8** *Page 22, Line 461: Maybe consider including such post-processing in this study. Using the bedrock mask can eliminate the fjord boundaries.*

▸ Thank you for this comment. We actually have considered this before submitting the paper but decided against it. This post-processing technique covers errors that the neural network makes (the coastline would not be detected as calving front if the rock would have been correctly recognized). If our models were used directly to produce new terminus delineations for further analysis, this post-processing would of course make sense and be highly beneficial. However, our models shall serve as baselines for future deep learning techniques and if a new model would enhance the recognition of rocks, this would not lead to an adequate decrease in the mean distance error compared to the baseline that uses this post-processing.

**2.9** *Table 8, Table 9, and Figure 9: The predicted terminus position for the Columbia glacier deviates from its true position, and the uncertainty for the Columbia glacier is too large.*

▶ Thank you again for this important point. As outlined before, the aim of this paper is not to provide a perfectly working deep learning model for calving front segmentation but to present the benchmark dataset for training and testing deep learning models as well as to provide baseline models to this dataset. Please refer to comment 1.1 and comment 1.2.

**Reviewer 2 - Major Concerns:**

**3.1** *It would be helpful to include the type of data covered by each study in Table 1, such what types of glacial features are provided by each dataset. These may include (but are not limited to) one or more of the following: full-line delineation (ESA), a centerline position (King), or glacial outline (GLIMS).*

▸ Thank you very much for the helpful recommendation! We updated Table 1 and its caption accordingly.

**3.2** *Since the primary data of interest is the training/testing benchmark dataset, it would be beneficial to further emphasize this in some way throughout the manuscript, such as including the total number of image pairs/metadata in the abstract. This will enable readers to see the primary contribution more easily, and enhance visibility/usage of this work within the field.*

▸ Thank you for this important suggestion. We revised our manuscript and put more emphasis on the benchmark dataset. Please see change 1, change 2, deletion 1, change 3, addition 1, change 4, change 5, addition 4, addition 9, addition 10, addition 11, and addition 3.

**Reviewer 2 - Specific Comments:**

**4.1** *P5 L108: "Lansat" -> "Landsat"*

▸ Thank you for this correction! You can find it in change 6.

**4.2** *P19 L364: "So there is a trade-off between patch size and batch size" Rephrase.*

▸ Thank you. We reformulated the sentence in change 7.

**4.3** *P24 L463: "are" -> "is"*

▸ Thank you. We integrated this correction in change 9.

**4.4** *P32 L539: "25rd" -> "25th"*

▸ Thank you for pointing out this mistake. We corrected it in the manuscript in change 10.

[revised manuscript text omitted]